**communications** engineering

# Hypersonic levitation and spinning: paving the way for enhanced single-cell analysis via contactless tissue dissociation
Yang Bai [1], Zhiwen Zheng[2,3], Zhaoxin Nie[1], Jialu Li [1], Zhihong Zhang [2] ✉ & Xuexin Duan [1] ✉

In the realm of single-cell analysis, effective tissue dissociation is a cornerstone yet often hampered by the drawbacks of traditional methods. Mechanical and enzymatic dissociation methods suffer from long processing times, reduced cell viability, and the loss of rare cell populations. Herein, we introduce a revolutionary tissue dissociation approach, Hypersonic Levitation and Spinning (HLS), which capitalizes on a uniquely designed triple-acoustic resonator probe. This probe enables the target tissue sample to levitate and execute a 'press-and-rotate' operation within a confined flow field, generating microscale 'liquid jets' that exert precise hydrodynamic forces in a non-contact manner. Through this mechanism, the shear forces on the tissue are enhanced, facilitating rapid and efficient dissociation while safeguarding cell integrity. We have further developed an automated tissue dissociation apparatus that integrates dissociation, fluid replacement, filtration, and output functions. Our comprehensive experiments on human renal cancer tissue dissociation, including flow cytometry, primary cell culture, immunofluorescence, and single-cell RNA sequencing, clearly demonstrate that Hypersonic Levitation and Spinning method not only greatly outperforms traditional techniques in tissue utilization (90% in 15 minutes vs. 70% in 60 minutes) and dissociation rate but also excels in maintaining high cell viability (92.3%) and preserving rare cell populations. This non-contact, gentle yet highly efficient dissociation method holds immense promise in diverse fields such as cell biology, single-cell sequencing, and precision medicine, expanding the scope of tissue dissociation technology and its applications.

Tissue dissociation is an essential process that underpins numerous biological and medical research applications. It serves as the gateway to isolating individual cells from complex tissues, thereby enabling a plethora of single-cell studies such as cell culture, flow cytometry, single-cell sequencing, rare cell isolation, and cell line development[1–4]. For instance, in cancer research, the ability to obtain single cells from tumor samples is crucial for identifying rare tumor cells that drive metastasis or exhibit drug resistance. This knowledge is fundamental for the development of targeted therapies and personalized medicine strategies[5–11]. In immunology, dissociating individual cells allows for the detailed characterization of diverse immune cell populations and their responses to pathogens or treatments[12–17]. Moreover, in broader biological research, scientists rely on tissue dissociation to uncover the variations in gene expression, protein levels, and metabolic activities among cells of the same type, which is vital for deciphering complex biological processes like development, differentiation, and disease progression[18–24]. Notably, single-cell research has emerged as a leading-edge technological approach, especially in cancer studies. It provides valuable perspectives on tumor heterogeneity, immune microenvironments, and therapy resistance, thereby deepening our comprehension of disease mechanisms.

However, the current state-of-the-art in tissue dissociation methods is far from ideal[25–29]. Traditional mechanical dissociation methods, while being simple and cost-effective, often inflict clear mechanical stress on cells[30]. This mechanical force, applied through techniques such as cutting, squeezing, mincing, and grinding, can lead to cell membrane damage, reduced viability, and even apoptosis. The results of these methods are highly operator-dependent and can vary considerably depending on the tissue type, thereby compromising the reproducibility of experiments[31,32]. Enzymatic methods,

[1]State Key Laboratory of Precision Measuring Technology and Instruments, Tianjin University, Tianjin, China. [2]Department of Urology, Tianjin Institute of Urology, The Second Hospital of Tianjin Medical University, Tianjin, China. [3]Department of Urology, The Second Affiliated Hospital of Anhui Medical University, Hefei, China. ✉e-mail: drzhihong@163.com; xduan@tju.edu.cn

on the other hand, utilize enzymes like collagenase, trypsin, dispase, and hyaluronidase to digest extracellular matrix components and intercellular junctions. Although they offer a more efficient dissociation process with higher cell viability and yield, they are not without drawbacks. Enzymatic digestion can be a time-consuming process, sometimes requiring several hours to achieve complete dissociation[33–37]. Additionally, careful optimization is necessary to preserve cell surface markers and functionality, as improper enzyme concentrations or digestion times can lead to the loss of important cellular characteristics.

Advances in micro- and nanotechnology have expanded the scope of tissue dissociation, especially for small tissues and single cells. These technologies offer precise control over the mechanical forces, reduce reagent usage, and enable automated workflows, which enhance the consistency of results[38,39]. For example, microfluidic devices have been developed to perform enzyme-free dissociation of stem cell aggregates or disperse neurospheres into individual cells[40–42]. These devices can integrate sample preparation with downstream applications, which is a clear advantage[43–47]. However, they are not without limitations. Microchannels in these devices can easily become obstructed by tissue fragments or cell aggregates, which restricts the flow and ultimately reduces the dissociation efficiency. Moreover, their limited tissue processing capacity makes them unsuitable for large-scale applications. Automated tissue grinders employ a motorized rotor and stator insert with microstructures to compress and break down tissue[48,49]. Although they offer faster treatment and improved reproducibility due to automation, the rigid contact between the 'teeth' and tissue samples still causes cell damage[50]. Additionally, the inability of the hard rotor and stator to adapt to the varying shape and size of tissue samples leads to uneven force distributions, which affects the system's performance and stability[51]. Furthermore, there have been many advancements in acoustic systems' manipulation at the micro- and nanoscale in recent years[52–56]. The advantage of acoustofluidic manipulation lies in its non-contact precision control, which allows for highly accurate manipulation of cells and particles without causing physical damage.

In light of these limitations, there is an urgent need for an ideal tissue dissociation tool that can overcome these challenges. Such a tool should be self-compatible with different types and sizes of samples, while simultaneously maintaining high cell viability and single-cell yields. This research gap has motivated our study, in which we introduce a contact-free tissue digestion method based on the Hypersonic Levitation and Spinning (HLS) approach. HLS capitalizes on the hypersonic streaming phenomenon, where a high-frequency acoustic wave interacts with fluid molecules to generate a steady flow of fluid. At GHz frequencies, the fluid particles move with high velocities due to the short decay length, allowing for precise control and manipulation of the fluid flow at a microscale[57–60]. By designing a hypersonic probe with a triple acoustic resonator, we are able to levitate the target tissue sample and perform a 'press-and-rotate' operation. This non-contact treatment not only applies instant hydrodynamic forces to the tissue with high precision but also enhances the shear forces during the self-spinning of the tissue. As a result, cell-cell and cell-matrix connections can be effectively disrupted, achieving rapid and thorough tissue dissociation while protecting cell integrity, increasing tissue utilization, and preserving rare cells. Our aim is to develop a method that not only addresses the limitations of existing techniques but also provides a more efficient and reliable solution for tissue dissociation. This, in turn, will lay a solid foundation for further exploration and advancement in precision medicine and related areas.

## Results and discussion

In this study, our primary objective was to develop a contact-free tissue dissociation method that could achieve high cell viability and yields. Cells within tissues are intricately organized in a highly specific manner, interacting through a complex web of mechanisms, including cell adhesion molecules, gap junctions, and the extracellular matrix. To achieve efficient transfer from solid tissue samples to individual cells, precise control of appropriate forces and conditions is essential. This ensures the successful dissociation and isolation of individual cells while preserving their viability and functionality.

The acoustic resonator serves as the conduit for applying acoustic energy, which is then transferred into the kinetic energy of a fluid via the hypersonic streaming effect. Under the constraints of boundary confinement, the Hypersonic Levitation and Spinning (HLS) approach engenders ordered quasi-static trapping conditions. In this context, the hydrodynamic force executes 'press-and-rotate' operations, leading to cell dissociation (Fig. 1a, see also Supplementary Information Fig. S1). The proposed HLS method integrates an acoustic resonator probe within a conical confinement structure. This combination not only enables the stable levitation of tissues but also induces their rapid self-rotation (Fig. 1b). The innovative design generates clear inertial effects that amplify the shear stress on tissue surfaces, expediting the dissociation process. Moreover, the detached cells revolve around the original tissue, effectuating an efficacious separation while maintaining their structural integrity. Beyond its mechanical role, hypersonic streaming also augments chemical processes, thereby disrupting cell-to-cell connections. This, in turn, enables the enzyme solution to permeate deeper into the tissue layers with greater facility. It thereby facilitates the binding of enzymes to collagen and matrix proteins within deeper cell connections, rupturing the most tenacious 'solid bridges' between cells and hastening enzymatic digestion.

To enable automatic tissue digestion, we also devised an automated tissue dissociation apparatus predicated on the HLS approach (Fig. 1c, see also Supplementary Information Fig. S2). This apparatus features four vertical tubes (designated for inlet, sample, outlet, and waste) and two chambers (for digestion and single-cell collection). This design seamlessly integrates dissociation and filtration, supplanting traditional labor-intensive procedures. It streamlines the process and augments the stability for contactless, automated single-cell preparation. The dissociation outcomes demonstrated by the HLS approach highlight its superior performance compared to conventional methods (Fig. 1d). Notably, the HLS approach accomplishes enhanced preservation of rare cell populations, encompassing fragile cell types. Additionally, it guarantees higher cell activity, a factor that is crucial for upholding the functional integrity and viability in downstream applications like primary culture and single-cell sequencing. The device also yields a substantially higher number of single cells within a shorter dissociation time, thereby attesting to its efficiency and its suitability for high-throughput biological research and clinical diagnostics. Collectively, these findings underscore the potential of the HLS technology as a transformative instrument in tissue dissociation, especially in scenarios where precision and minimal cell loss are imperative.

## Probe design

Based on the tissue dissociation goals mentioned previously, the key to achieving noncontact, high-speed tissue dissociation through HLS lies in the coordinated design of the acoustic resonator probe and the confinement structure. This allows the tissue block to remain stably suspended and spinning at high speed. To assist the experiment in selecting suitable device positioning, height, cone angles, and other parameters, we initially simulated a confined hypersonic streaming field. To simplify the model and numerical calculations, no elastic materials were included in the simulation. As depicted in Fig. 2a, the high-intensity hypersonic streaming velocity field generated by the GHz resonator propels the liquid forward, creating a hypersonic streaming jet with a maximum speed reaching the order of meters per second (m/s). Under the confinement of the conical structure, the hypersonic streaming jet progressively converges towards the base, generating a strong upward lift. Beyond the convergence effect, the conical base also reflects part of the stream upwards due to the boundary conditions, further enhancing the upward fluid lift. The tissue block achieves suspension within the liquid through a balance of gravity, buoyancy, and fluid lift. Notably, at an asymmetric spatial placement (e.g., the position of the resonator to one side of the conic), the hypersonic streaming can exert greater pressure on one side of the tissue block. The generated eccentric pressure (Fr) and shear force (Fs) together create an eccentric torque that

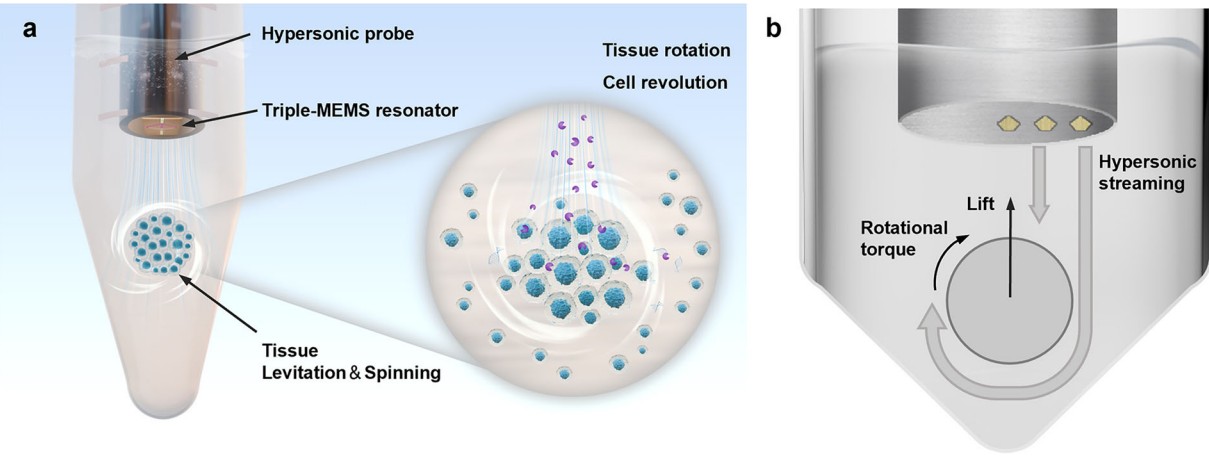

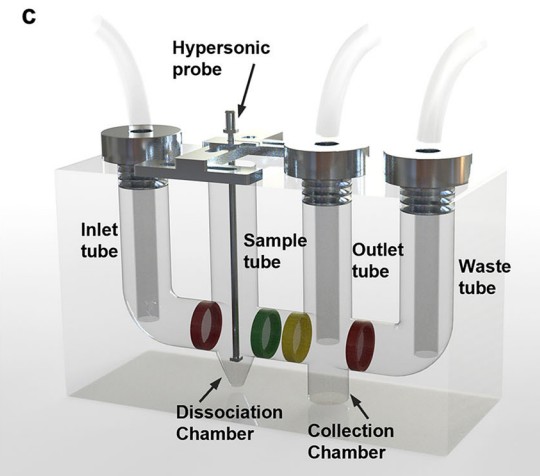

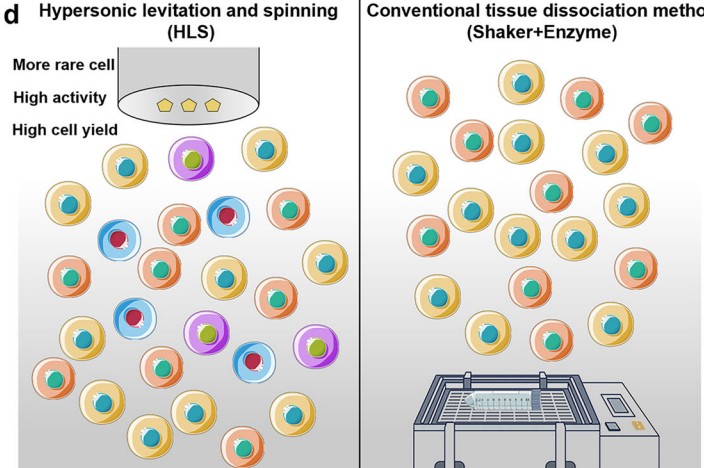

**Fig. 1 | Overview of the HLS tissue dissociation strategy. a** Schematic of the Hypersonic Levitation and Spinning (HLS) principle. **b** Structure of the triple-acoustic resonator probe and the hydrodynamic force it generates via hypersonic streaming. **c** Illustration of the automated tissue dissociation apparatus integrating dissociation and filtration functions. **d** Dissociation outcomes showing improved rare cell preservation, enhanced cell activity, and increased cell yield achieved by the HLS method.

initiates the target spinning. The conical structure further guides the streaming at the base of the tissue block to form a vortex that generates shear forces aligned with the direction of the eccentric torque. The combined effect of these torques results in rotational torque, ensuring that the tissue block achieves stable, high-speed spinning in suspension.

Moreover, in this rotational flow field, a centripetal force induced by the velocity gradient also plays a role. Tissue fragments of appropriate size can be stably trapped at the center of the vortex and undergo self-rotation. Under the combined action of pressure, shear force, and centripetal force, enzymes penetrate more rapidly into the deeper layers of the tissue. As a result, larger tissue blocks gradually break down into smaller fragments, which then transition to orbital motion along the vortex flow. The entire dissociation process is governed by the complex interplay of eccentric thrust (Fr), rotational shear (Fs), and centripetal force, which collectively accelerate enzymatic penetration and tissue dissociation.

Among the three resonators, the first and second resonators primarily generate hypersonic streaming that applies constant eccentric pressure (Fr) on the surface of the tissue block. The third resonator works in coordination with the conical structure, creating a swirling vortex that exerts shear forces (Fs) on the tissue while inducing high-speed rotation. Therefore, the design of the triple resonator enables simultaneous pressure and rotational effects, which cannot be achieved with a single resonator (Fig. 2b). This configuration of the acoustic flow field enhances the driving capability for the tissue, with the streaming covering the entire conical region and ensuring the stability of tissue rotation.

In this study, Fr and Fs are quantified by integrating the pressure and shear stress over the tissue surface, respectively. Mathematically, the net pressure force Fr can be expressed as

$$Fr = \iint_s p(x)dA \approx \iint_s \frac{1}{2}\rho v(x)^2 dA \qquad (1)$$

where p(x) is the local pressure (including dynamic pressure) acting on the tissue surface, ρ is the fluid density, and v(x) is the fluid velocity.

The shear force Fs arises from the velocity gradient at the tissue–fluid interface and is given by

$$Fs = \iint_s \tau(x)dA = \iint_s \mu \frac{v_t}{n} dA \qquad (2)$$

μ is the fluid viscosity, $v_t$ is the tangential fluid velocity component at the surface, and ∂$v_t$/∂n is the velocity gradient in the direction normal to the surface. In the subsequent simulations and force-measurement experiments, we optimized parameters to enhance Fr and Fs.

This synergistic effect between the hypersonic streaming jet and the conical structure was further analyzed through simulations of different cone angles, as shown in Fig. 2c. According to the simulation, cone angles between 30 and 70 degrees can ensure fluid lift and rotation of the tissue block. However, a 30-degree cone angle creates excessive fluid convergence, resulting in an overly narrow levitation and rotation area, while a 70-degree

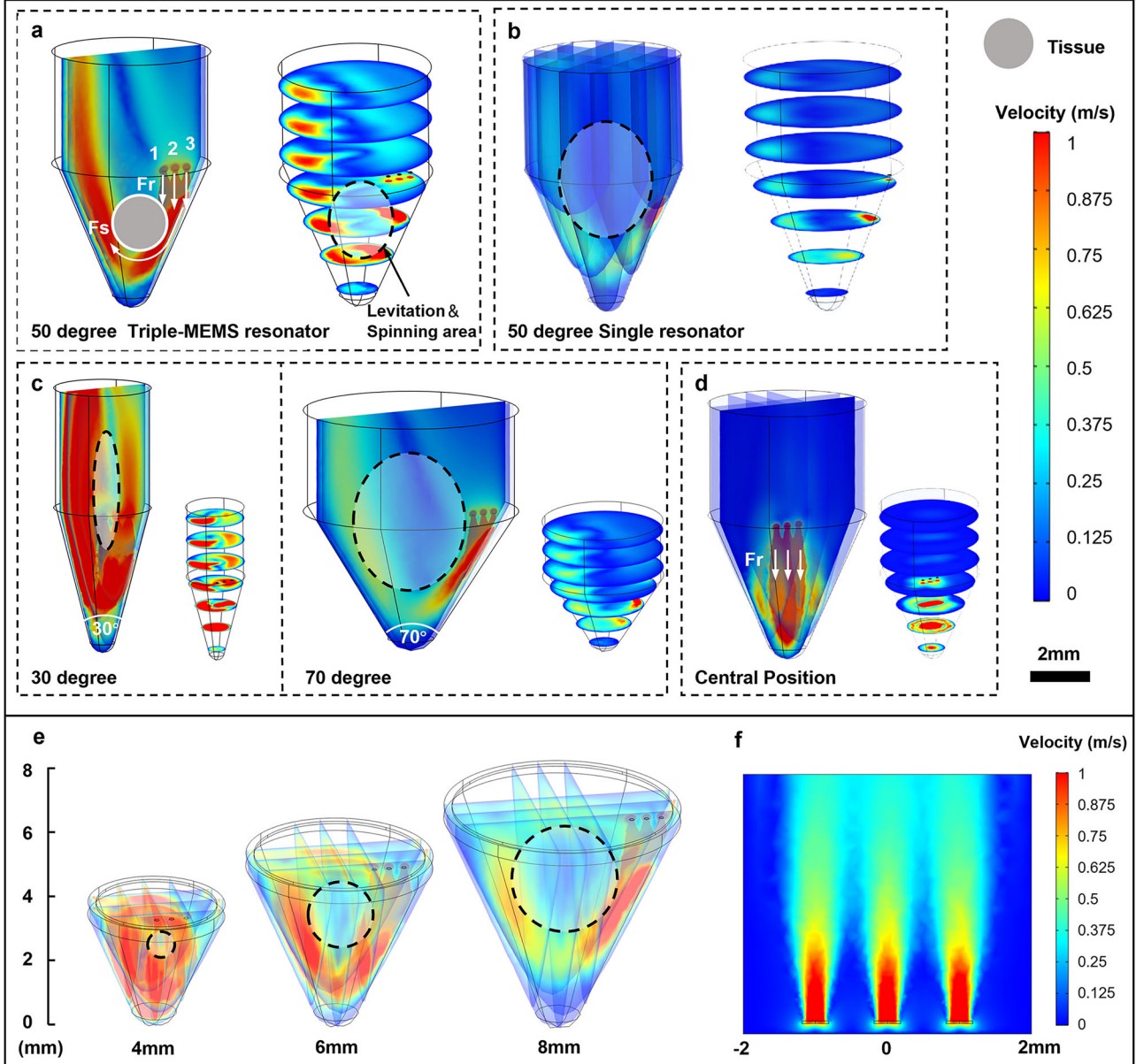

**Fig. 2 | Simulation analysis of hypersonic streaming in the HLS system. a** Flow field simulation of the triple-resonator structure with a 50° conical base. **b** Flow field simulation of the single-resonator structure with a 50° conical base. **c** Effect of different conical angles on the hypersonic streaming. **d** Flow field simulation when the device is positioned at the center of the conical tube. **e** Effect of probe height on the streaming pattern. **f** Flow velocity distribution generated by the triple-resonator configuration. Fr represents the eccentric pressure; Fs represents the shear force; 1, 2, and 3 denote the three resonators on the triple-resonator probe.

cone angle weakens fluid convergence, making the levitation and rotation area too broad to sustain tissue self-rotation. A 50-degree cone angle achieves optimal convergence, generating a strong vortex suitable for effective tissue dissociation. This configuration enhances the velocity gradient of the vortex flow, thereby substantially boosting shear forces acting on the tissue, facilitating efficient tissue dissociation while maintaining cell integrity. Additionally, this angle closely resembles the shape of standard centrifuge tubes, ensuring compatibility for accelerated dissociation in standard laboratory equipment. In addition, we performed simulations with the device placed at the exact center of the structure, as shown in Fig. 2d. It can be seen that when the device is positioned at the center, the generated jet fails to form recirculating vortices after impinging on the conical base, and thus cannot induce tissue rotation. This explains why the device was placed in an off-center position.

Through simulations, we optimized the resonator height to ensure stable tissue spinning and levitation (Fig. 2e). Adjusting the height alters the

size of the levitation and spinning area, which must be appropriately matched to the tissue size for effective dissociation. If the height is too low, the tissue can become stuck and fail to rotate. Conversely, if the height is too high, the tissue may rotate around the axis with the vortex flow, reducing the shear forces due to the increased spatial distribution. For tissue sizes ranging from 1 to 3 mm³, a resonator height of 6 mm was found to be the most suitable. At this height, the tissue can levitate and rotate stably at the vortex center, where the velocity gradient on the tissue surface is maximized, resulting in the highest shear forces. This balanced levitation and spinning area ensures stable tissue self-rotation and facilitates efficient dissociation.

Considering that the typical tissue size used in the dissociation experiments is approximately 1 to 3 mm³ and that the size of single resonator ranges from $1 \times 10^4$ µm² to $1 \times 10^5$ µm², we set the spacing between each resonator at 1 mm to ensure sufficient liquid flow coverage. At such configuration (Fig. 2f), the three resonators in this probe can work individually without interfering with each other.

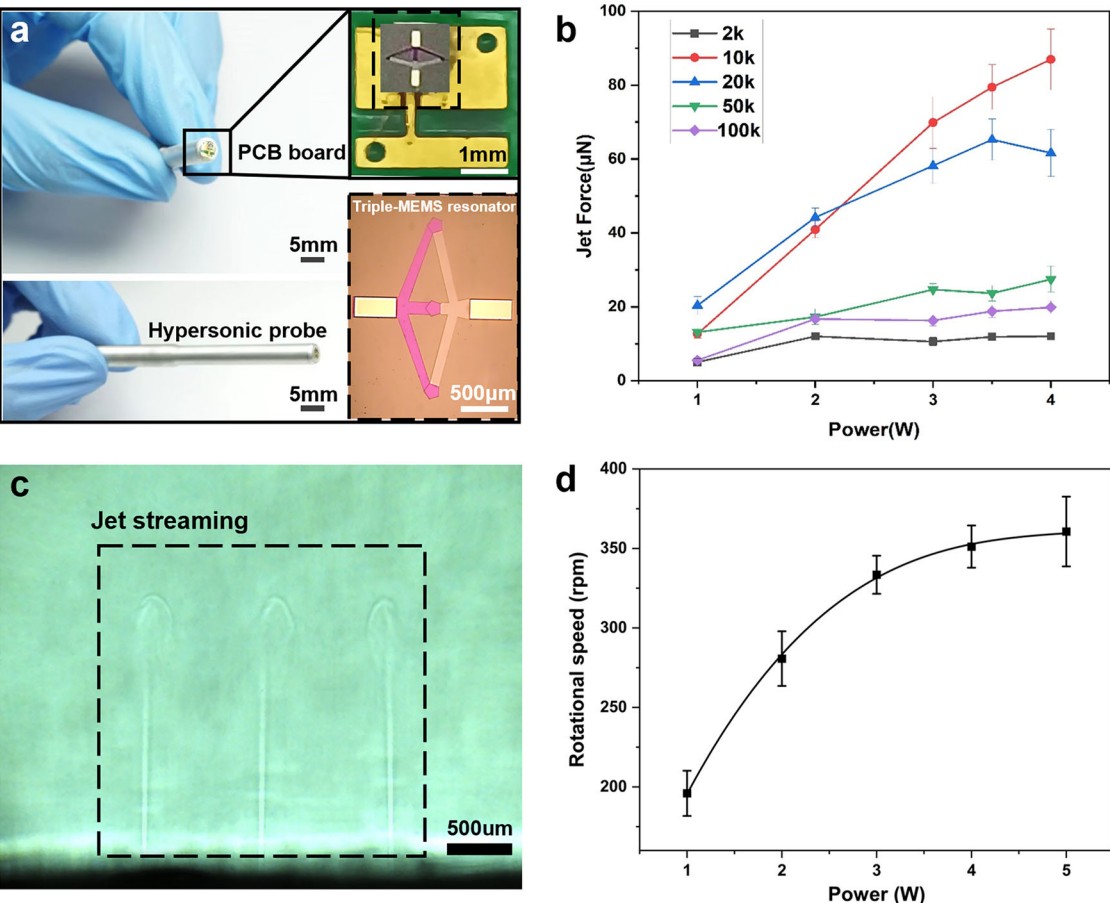

**Fig. 3 | Characterization of the triple-MEMS resonator probe and its acoustic jetting performance. a** Optical images of the triple-MEMS resonator probe and its packaging. **b** Jetting force of resonators with different sizes under various input powers. **c** High-speed microscopy images showing jet streams generated by the triple-resonator probe. **d** Rotation speed of the tissue sample under different power levels. Error bars represent mean ± standard deviation from $n = 3$ independent samples.

## Probe fabrication and hypersonic streaming characterization

The fabrication of the probe is compatible with the standard MEMS approach. The fabricated probe and its packaging are presented in Fig. 3a. A parallel on-chip circuit design was adopted to evenly distribute power across the three resonators. The characteristic impedance was meticulously optimized to ensure a balanced power input. This uniform power distribution is crucial as it enables each resonator to generate an identical jet force, thereby precluding any instability in tissue rotation. To facilitate the proper fitting of the probe and its efficient alignment with the dissociation chamber, the probe resonator was encapsulated within a steel wand with a 5 mm inner diameter. The micrograph showcases the compact arrangement of the resonators, which is fundamental for the precise control of the hypersonic streaming flow field (Fig. 3a). The structural diagram of the probe is illustrated in Supplementary Information Fig. S3a.

To further refine the power parameters of the triple-resonator system, a series of mechanical characterization experiments were conducted. We employed a miniaturized force sensor (Aurora Scientific, 405 A) to measure the jet force and thereby optimize the power and size selection of the individual resonators. The force sensor probe was precisely positioned at the center of the acoustic resonator, maintaining a fixed distance of 100 μm (as shown in Supplementary Information Fig. S3b). Jet forces were measured across five distinct device sizes and under diverse power settings. As depicted in Fig. 3b, within a relatively lower power range, the jet force and input power exhibited an approximately linear relationship. However, at higher power levels, the force tended to saturate. Moreover, excessive power generation leads to heat production, which can irreversibly damage cells. Consequently, the power settings were prudently limited to below 3 W to

ensure both safety and efficiency. Based on the force measurements, resonators with an area of 10k (10000 μm²) were selected due to their capacity to generate sufficient jet forces (ranging from 10 to 70 μN) within the 1–3 W power range. This force range is effective in driving tissue dissociation without inducing cell damage[61,62]. Additionally, the 10k resonators demonstrated the most favorable linearity in force output across this power range, thus guaranteeing stable and predictable performance during operation.

The jet streaming from the triple-resonator probe is illustrated in Fig. 3c. A triple parallel jet flow was distinctly observed, which clearly indicates the uniform distribution of the jet forces. This result not only validates the fabrication quality of the device but also aligns well with our initial design (Fig. 2f). Unlike traditional dissociation methods, where the tissue dissociates by moving with the fluid flow (translational motion), in the HLS system, more of the fluid energy is utilized to rotate the tissue rather than letting it move along with the fluid. The self-rotation of the tissue in the HLS system generates stronger shear forces. These amplified shear forces, resulting from the tissue's rotation around its own axis, are more effective at stripping dissociated material and disrupting cell-cell and cell-matrix connections, thereby accelerating the enzymatic digestion process. Tissue rotation characterizations were carried out under different power levels using a typical tissue sample of mouse kidney tissue. The results, as presented in Fig. 3d, demonstrate that the tissue rotational speed escalates with increasing power and reaches a saturation point at higher power levels. The fitted curve indicates that the system attains a rotational speed of up to 5.5 rps at 3 W, thereby providing an optimal balance between dissociation efficiency and power consumption. Notably, this trend is in concordance

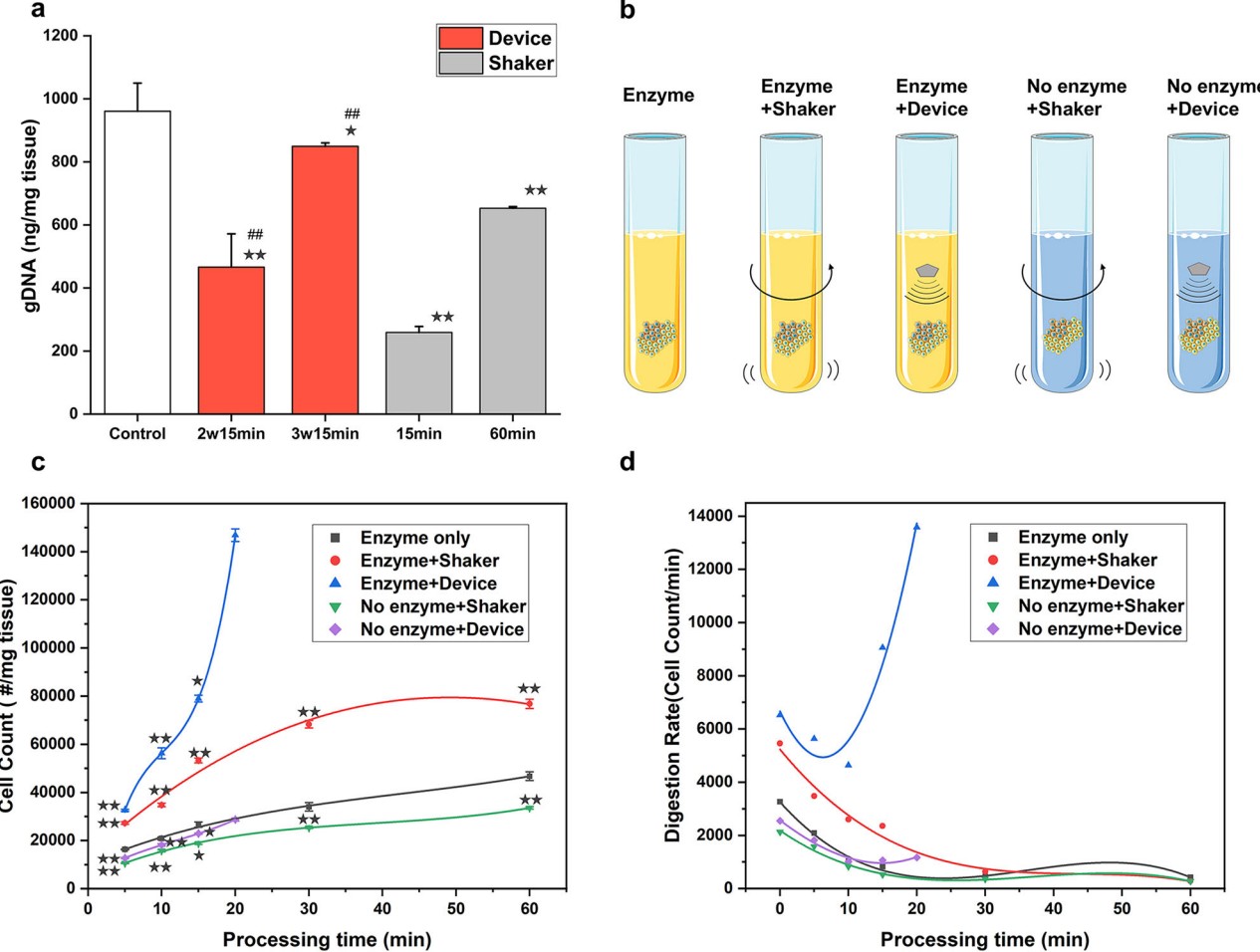

**Fig. 4 | Comparison of tissue dissociation efficiency using HLS and conventional shaker-based enzymatic methods. a** Quantification of tissue yield using genomic DNA content. "2w15min" and "3w15min" indicate treatment with the device at 2 W and 3 W power for 15 min, respectively. **b** Experimental groups used for characterizing dissociation rate. **c** Fitted curves showing changes in single-cell yield over time in different groups. **d** Dissociation rate curves over time for each experimental condition. A two-sided t-test was used for statistical testing. Stars indicate $p < 0.05$ and double stars indicate $p < 0.01$ relative to the control. Cross-hatches indicate $p < 0.05$ and double cross-hatches indicate $p < 0.01$ relative to the static condition at the same digestion time. Error bars represent mean ± standard deviation from $n = 3$ independent samples.

with the force curve shown in Fig. 3b, where 3 W power utilization exhibits the highest efficiency. Beyond this power level, the marginal increase in force results in a negligible improvement in tissue rotational speed. Supplementary Movie S1 offers additional visual insights into the tissue rotation process under these conditions.

Therefore, at a power of 3 W, the system achieves the peak power utilization efficiency, generating greater fluid velocity and force compared to lower power levels. This augmentation of fluid forces, encompassing both Fr and Fs, plays a pivotal role in enhancing the dissociation rate.

**Tissue dissociation**

Tissue dissociation is a crucial aspect of the efficacy of our proposed method, as it directly relates to the overall goal of obtaining high-quality single-cell suspensions. The tissue utilization rate, which represents the proportion of dissociated tissue relative to the total tissue volume, is a key metric for assessing the processing capability of the device[38]. A higher utilization rate implies that a greater amount of tissue is effectively dissociated into single cells, thereby enhancing the efficiency and data yield of downstream applications.

To test the tissue dissociation efficacy of our device after optimization, we conducted experiments using a sample of human renal cancer tissue. Equal-weight tissue samples were prepared and processed for 15 min using the HLS method. Subsequently, the samples underwent collection, filtration,

elution, and genomic DNA (gDNA) extraction procedures. In the control group, the tissue was minced and immediately placed into a reagent kit for gDNA extraction to establish the upper limit of gDNA recovery.

At an average power of 2 W, the gDNA yield after 15 min was approximately 50% of that of the control group. However, when the average power was increased to 3 W, the yield substantially increased to approximately 90% of that of the control (Fig. 4a). In contrast, the traditional shaker method, which was carried out at 37 °C and 600 rpm for one hour, yielded gDNA at approximately 70% of the control. These results suggest that prolonged soaking in enzymatic solution, as in the case of the traditional method, may lead to damage to the surface proteins of dissociated cells, consequently reducing cell viability. Therefore, minimizing the dissociation time is of utmost importance for preserving cellular integrity.

Our experimental findings clearly demonstrate that the HLS method achieved a remarkable tissue utilization rate of 90% within just 15 min. In contrast, the traditional method only achieved a rate of 70% after 1 h of dissociation. Evidently, the hypersonic streaming dissociation technique employed by HLS substantially enhances the dissociation efficiency, enabling greater tissue utilization in a clearly shorter time compared to conventional methods.

Next, we examined the dissociation rate in the same renal cancer tissue. The experimental conditions and grouping are shown in Fig. 4b. The statistical results (Fig. 4c) revealed that the cell counts in the enzyme + HLS

group at identical time intervals were considerably higher than those of the other groups, maintaining the highest dissociation rate throughout the process. The no-enzyme + HLS group exhibited a lower dissociation rate, which was comparable to that of the enzyme-static group and slightly higher than that of the no-enzyme + shaker group. These observations confirm two important findings: (1) The hypersonic streaming dissociation utilized by HLS gently accelerates the enzymatic process, enhancing the interaction between the enzyme and the tissue without causing clear alterations in tissue or cell morphology. This leads to an increased rate of enzymatic digestion, facilitating both efficient and rapid dissociation. (2) Regardless of the addition of enzyme solution, the dissociation rate with HLS is consistently higher than that of the shaker group. This highlights the superiority of hypersonic streaming over traditional methods in enhancing fluid dynamics and enzymatic reactions.

After 10 min of dissociation, a notable increase in the dissociation rate was observed specifically in the enzyme + HLS group, which was distinct from the other groups (Fig. 4d). This phenomenon can be attributed to the rotation and dissociation of large tissue blocks under the influence of hypersonic streaming prior to the 10-min mark. During this process, the tissue is suspended and rotates within the levitation and spinning area, allowing for a gradual dissociation from the outer layers toward the inner core. As the dissociation process progresses, the tissue blocks become looser. After approximately 10 min, the larger tissue blocks break down into several smaller fragments, thereby increasing the total surface area and exposing more substrates for enzymatic binding. This results in an increasing trend in the dissociation rate after 10 min, a trend that was not observed in the other groups, which may have failed to sufficiently disperse the tissue. This suggests that the unique tissue dispersal mechanism enabled by hypersonic streaming is a key factor contributing to the increased dissociation rate observed with HLS.

## Dissociated cell characterization

Primary cell culture is a cornerstone in biomedical research, yet it is fraught with challenges. The success of primary cell culture hinges on the quality of the starting cell population, which is directly influenced by the tissue dissociation method[48–51]. The ability to obtain single-cell suspensions with high viability and minimal damage is essential for elucidating cellular mechanisms and advancing therapeutics. It serves as a fundamental platform for studies on carcinogenic factors, in vitro drug screening, and regenerative organ studies. However, achieving optimal dissociation while maintaining cell integrity is a delicate balance that requires a highly efficient method.

To assess whether the single-cell suspensions obtained via our HLS device meet the stringent criteria for primary cell culture, we conducted primary cell cultures using single-cell suspensions dissociated from a surgical resection sample donated by a kidney cancer patient. Kidney cancer is a complex disease with heterogeneous cell populations, making it an ideal model to evaluate the performance of our dissociation method in handling diverse cell types. Moreover, the ability to obtain viable and functional single cells from kidney cancer tissue is crucial for understanding the pathophysiology of the disease and developing targeted therapies.

A 100 mg sample of human kidney cancer tissue was minced and divided into two equal parts of 50 mg each. One portion was placed in the dissociation chamber of the HLS device, along with an adequate amount of collagenase II solution. The device was then activated at 3 W for 15 min. This specific power and time combination was determined based on our previous optimization experiments, which demonstrated that these parameters yielded the best results in terms of tissue dissociation and cell viability. After the dissociation process, the single-cell suspension was collected via the device's internal filter.

The other portion of the tissue was placed into a centrifuge tube and digested with the same concentration of enzyme mixture by shaking on a shaker for 60 min. This traditional shaker method was used as a control to compare the performance of our HLS device. The longer digestion time for the shaker group was chosen because previous studies have shown that it typically requires a more extended period to achieve comparable

dissociation using this method. After digestion, the cell suspension was filtered through a cell strainer.

Both groups of collected cell suspensions were cultured under identical conditions in cell culture incubators. Cells were grown in Dulbecco's modified Eagle medium (DMEM) supplemented with 10% fetal bovine serum and 1% penicillin–streptomycin. The cell line was maintained in T-25 cell culture flasks in an incubator at 37 °C and 5% $CO_2$. Daily photography observation was carried out over a period of 11 days, with timely medium changes to ensure a conducive environment for cell growth. Every other day, cell growth was monitored at five fixed locations using an inverted microscope to capture images and observe the cell proliferation and morphology.

As illustrated in Fig. 5a, cells from the HLS device group exhibited more initial cell adhesion and slightly higher overall proliferative efficiency compared to the shaker group. The additional images provided in the Supplementary Information Figs. S4 and S5 further support this observation. From the cell counting results shown in Fig. 5b, the device group presented an average of 50% more adherent cells than the shaker group over 11-day primary culture. The growth rate of adherent cells every two days (Fig. 5c) revealed that the cell growth rate of the device group was 20% greater than that of the shaker group during the first cycle, indicating faster and greater initial cell adhesion. The more efficient dissociation method employed by the HLS device, which causes minimal cell damage, likely facilitates cell adhesion. For each subsequent cycle, the growth rates of both groups were nearly identical, suggesting that cell viability after device dissociation remained unaffected compared with traditional methods, thereby supporting regular cell division. These results strongly suggest that, compared with traditional dissociation methods, the HLS dissociation device preserves cell surface antigens and proteins during the dissociation process and maintains cell viability, enabling faster cell adhesion and growth. This, in turn, can greatly improve the success rate of primary cell culture, which is of great importance for downstream applications.

To further demonstrate the membrane protein activities of the dissociated cells, we conducted immunofluorescence testing using the above primary cultured kidney cancer cells dissociated by the HLS device. Confocal microscopy images (Fig. 5d) show that the CA9 sites on the cell membrane were successfully stained red and that CD10 was stained green. The staining was clear and distinct, indicating that the cells can be clearly identified as renal cancer cells. This result is consistent with the pathology test results. Therefore, the cell surface antigens remain intact after treatment with the HLS dissociation device, permitting immunofluorescence. This finding also indicates that the fast dissociation speed and short duration characteristics of the HLS can be highly advantageous for quick clinical testing and preliminary disease screening, especially for biopsy examination during surgery. The ability to rapidly obtain viable and identifiable cells with intact membrane proteins is crucial for timely and accurate diagnosis, highlighting the practical value of our method in a clinical setting.

## Dissociation time-dependent flow cytometry analysis

Flow cytometry is a powerful technique that allows for the detailed characterization of cell populations based on their physical and chemical properties. In the context of our study, we utilized flow cytometry to gain a deeper understanding of the hypersonic streaming-mediated cell dissociation process, with a particular focus on the dynamic changes that occur over time.

The ability to control the dissociation of cells over time is a unique feature of our HLS method, and it provides a valuable tool for investigating the cell arrangement in tissue. By varying the treatment time, we can observe how different cell populations are released from the tissue at different stages of the dissociation process. This temporal resolution is crucial for unraveling the complex interactions between cells and the extracellular matrix, as well as for understanding the mechanisms underlying cell dissociation.

To this end, we conducted flow cytometry analysis on dissociated cells after different treatment times using clear cell renal carcinoma tissue. We labeled cell suspensions with antibodies and fluorescent probes specific for CA9 (renal cancer cells), CD45 (leukocytes), and 7-AAD (live/dead),

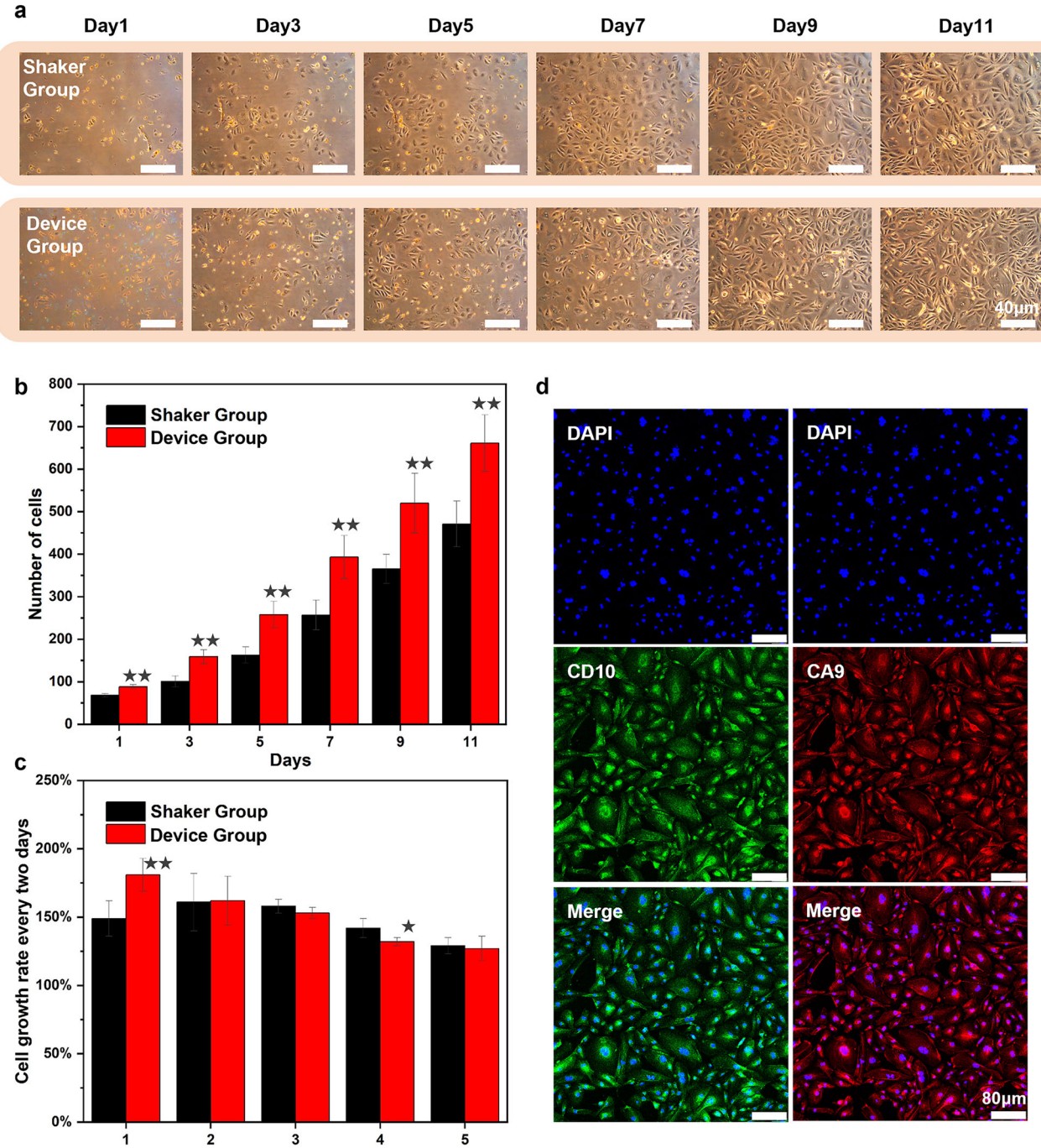

**Fig. 5 | Evaluation of cell growth and phenotypic markers following tissue dissociation. a** Representative cell growth images from the same location over an 11-day primary culture period in the shaker group and HLS group. All scale bars represent 40 μm. **b** Quantification of cell numbers and growth rates across the 11-day culture period. **c** Cell growth rate for each cycle (one cycle = 2 days). **d** Immunofluorescence characterization of dissociated human renal cancer tissue, showing clear cell renal carcinoma (CA9⁺, CD10⁺). All scale bars represent 80 μm. DAPI (4',6-diamidino-2-phenylindole), a fluorescent nuclear stain, was used throughout the experiments. A two-sided t-test was used for statistical testing. Stars indicate $p < 0.05$ and double stars indicate $p < 0.01$ relative to the control. Error bars represent mean ± standard deviation from $n = 5$ independent image-based measurements.

enabling us to classify the cells into three distinct categories based on their expression: CA9 + CD45 - (cancer cells), CA9 - CD45+ (immune cells), and CA9 - CD45 - (stromal cells) (see Supplementary Information, Supplementary Table S1).

Our results revealed fascinating insights into the dissociation process. Notably, there were variations in the proportions of cells with different dissociation times (Fig. 6a). For cancer cells, the proportion in the HLS (5 min) group was the highest at 56%, decreased to 33% in the HLS (10 min) group, and then increased again to approximately 50% in the HLS (15 min) group, slightly exceeding that of the control (40%). This dynamic pattern of cancer cell dissociation suggests a complex process of tissue breakdown. Initially, the outer layers of cancer cells are likely to be preferentially dissociated, leading to a higher proportion of cancer cells in the early stage. As the dissociation progresses, deeper layers of the tissue are disrupted, releasing additional cancer cells that were initially enclosed within the tissue structure. This phenomenon was further supported by our previous tissue

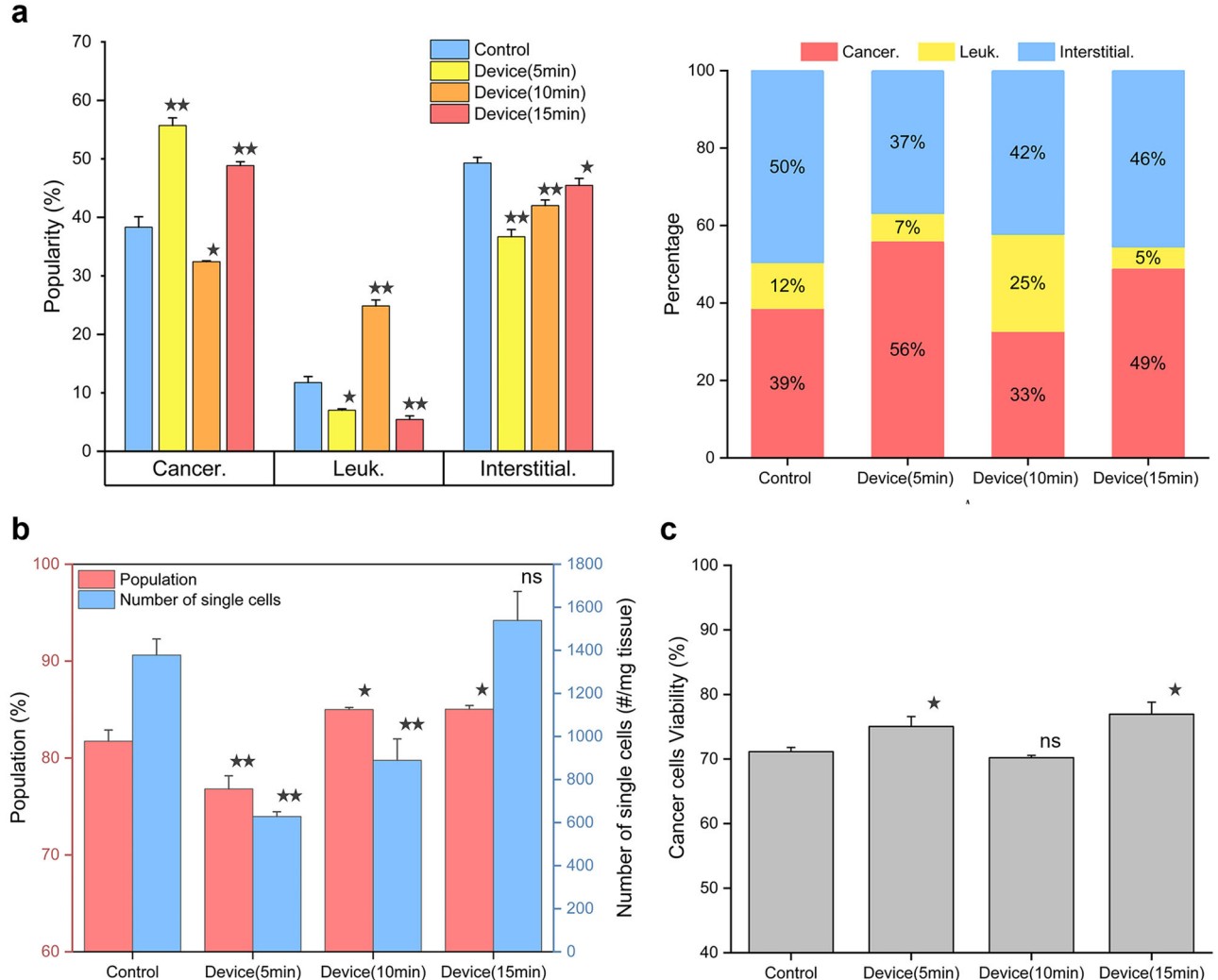

**Fig. 6 | Flow cytometry analysis of cells dissociated from human renal cancer tissue. a** Proportions of different cell types identified by flow cytometry. **b** Statistical comparison of single-cell proportions and absolute counts across experimental groups. **c** Cancer cell viability statistics for each group. Stars indicate $p < 0.05$ and double stars indicate $p < 0.01$ relative to the control. Error bars represent mean ± standard deviation from $n = 3$ independent samples.

utilization characterizations, where the HLS (15 min) group achieved a 90% tissue utilization rate compared to the 70% utilization rate in the control group. The difference in tissue utilization indicates that there may be a substantial number of cancer cells remaining in the tissue in the control group, which could account for the observed changes in cancer cell proportions over time.

In contrast, immune cells peaked in the HLS (10 min) group at 25%, which was 2–4 times greater than that in the other groups. The increase in immune cell release at this time point may be related to the disruption of the tumor microenvironment (TME). As the dissociation process affects the tissue architecture, immune cells that are associated with the cancer cells or located in the vicinity may be released more readily. This finding provides valuable information about the spatial distribution of immune cells within the tumor and their interactions with cancer cells during the dissociation process. Moreover, this suggests that the HLS method facilitates the early release of immune cells from the tissue, potentially indicating that traditional dissociation methods may incompletely dissociate tissues, leading to an incomplete understanding of the tumor immune microenvironment.

The proportions of stromal cells were relatively similar across all the groups, with the percentages in the HLS (15 min) and control groups being approximately 50%, which was slightly greater than those in the other groups. The relatively stable proportion of stromal cells suggests that they may be more resistant to dissociation compared to cancer and immune cells,

or they may be released in a more gradual manner throughout the dissociation process.

Further analysis of cell proportions over time revealed a clear dynamic trend. Cancer cell proportions initially decreased but then increased, while immune cells showed an inverse pattern, peaking at 10 min. The number of stromal cells consistently increased over time. This trend can be attributed to the hierarchical organization of the tumor tissue, where cancer cells are embedded within layers of stroma, blood vessels, and immune cells. As the dissociation progresses, the outer layers of cancer cells are dissociated first, followed by the release of immune cells and the gradual breakdown of the stroma. The observed increase in immune cell proportions between 5 and 10 min may be due to the dissociation of surrounding cancer cells, which exposes and releases the immune cells that were previously in contact with or adjacent to the cancer cells. The continuous increase in stromal cell numbers over time indicates that the dissociation of stromal components is a progressive process that contributes to the loosening of the tumor structure and the release of more cells.

In addition to analyzing the dissociation process, we also assessed the single-cell rates and cell viability through flow cytometry. The results showed that the HLS device group achieved a faster dissociation rate and a higher yield of single cells compared to the control group, which is consistent with our previous observations. As illustrated in Fig. 6b, the control group achieved an 82% single-cell rate after 60 min of dissociation, while the

HLS group presented a 77% single-cell rate after 5 min. With extended dissociation times, the HLS (10 min) and HLS (15 min) groups achieved single-cell rates exceeding 85%, surpassing the control group. This indicates that although shorter dissociation times may result in incomplete dissociation of cell clusters, the single-cell rate stabilizes above 85% as the dissociation progresses. The HLS device yielded 1600 single cells/mg in 15 minutes, compared to 1400 single cells/mg achieved by the control after 60 min, highlighting the device's efficiency in achieving single-cell dissociation in a shorter time frame. The larger increase in single-cell numbers from 10 to 15 min compared to that from 5 to 10 min is also consistent with our earlier dissociation rate observations, suggesting that the breakdown of larger tissue fragments into smaller pieces after 10 min facilitates the transition of cell clusters into single cells. Some discrepancies between flow cytometry and dissociation rate data may be attributed to cell loss during staining, washing, filtering, or differences in tissue type, which are common challenges in cell analysis techniques.

Regarding cell viability, flow cytometry indicated a 70% viability rate for cancer cells in the control group, which was similar to the HLS (10 min) group. The HLS (5 min) and HLS (15 min) groups showed slightly higher viability rates of around 75% (Fig. 6c). The shorter dissociation time in the HLS (5 min) group may help preserve cell viability by reducing the exposure of cells to enzymatic solutions and mechanical stress. In the HLS (15 min) group, the higher viability rate despite the longer dissociation time may be due to the gentle nature of the hypersonic streaming, which minimizes damage to cells. These findings suggest that the hypersonic streaming-based dissociation method can achieve efficient single-cell generation while maintaining cell integrity, offering great potential for a wide range of applications in both research and clinical settings.

Overall, the dissociation time-dependent flow cytometry analysis using our HLS method has provided deeper insights into the cell dissociation process. The ability to monitor the dynamic changes in cell populations over time has allowed us to uncover the complex interactions within the tumor microenvironment and the mechanisms underlying tissue dissociation. These findings not only enhance our understanding of the biology of tumor tissue but also have important implications for the development of more effective dissociation methods and the improvement of single-cell analysis techniques.

## Single-cell RNA sequencing

Single-cell RNA sequencing (scRNA-seq) has become a powerful tool in the life sciences, enabling high-resolution analysis of cellular heterogeneity. The quality of scRNA-seq data is highly dependent on the quality of the single-cell suspensions used as input. Undissociated tissue or cell damage resulting from suboptimal dissociation methods can lead to the loss of certain cell types, particularly rare cells, and thus compromise the accuracy and comprehensiveness of the analysis. In this context, our HLS method, which has demonstrated superior tissue utilization, dissociation rate, and cell viability compared to traditional methods, holds great promise for improving the quality of single-cell suspensions for scRNA-seq.

To investigate the potential of the HLS method in preserving the integrity of tissue for scRNA-seq analysis, we performed scRNA-seq experiments using human renal cancer tissue samples. The samples were carefully divided into experimental and control groups, ensuring equivalent tissue masses to minimize potential discrepancies in dissociation results. The experimental group was dissociated using the HLS device for 15 min at 3 W, a parameter optimized based on our previous studies to balance dissociation efficiency and cell viability. In contrast, the control group underwent dissociation via a shaker for 60 min, representing a traditional enzymatic dissociation method.

After dissociation and filtration, single-cell suspensions were concentrated for sequencing. The results were striking. The HLS group achieved a cell concentration of $9.37 \times 10^5$ cells/mL with a remarkable viability of 92.3%, while the control group yielded $8.76 \times 10^5$ cells/mL with 87.2% viability (Fig. 7a). These findings not only confirm the higher cell recovery and viability of the HLS method within a shorter dissociation time but also

suggest that the HLS method may have a gentler impact on cells, as evidenced by the higher viability. This is consistent with our previous observations on tissue utilization and cell dissociation, further highlighting the advantages of the HLS approach. Additionally, the average cell size in the HLS group was 14.9 μm, compared with 13.2 μm in the control group. This difference may indicate that the HLS method is more effective in retaining a broader range of cell sizes, potentially reflecting a more comprehensive preservation of the original tissue's cellular composition. This is of particular significance as it suggests that the HLS method may be better able to capture the full spectrum of cell types, including those that may be more sensitive to mechanical or enzymatic stress.

Both single-cell suspensions were processed using a 10X Chromium platform, yielding a total of 34,008 cells sequenced (19,860 from the HLS group and 14,148 from the control group). The scRNA-seq quality metrics were comparable between the two groups, indicating that the HLS method did not compromise the quality of the sequencing data. Clustering and annotation of the sequenced cells revealed 11 distinct cell clusters, encompassing endothelial cells, plasma cells, renal carcinoma cells, smooth muscle cells, macrophages, mast cells, and lymphocytes (Fig. 7b). The relative proportions of these cell clusters were similar across both groups (Fig. 7c), suggesting that the HLS method is capable of preserving the overall cellular composition of the original tissue without introducing substantial biases during dissociation. This is a crucial finding as it validates the reliability of the HLS method in maintaining the integrity of the tissue's cellular architecture.

However, a closer examination of the rare cell clusters revealed a substantial advantage of the HLS method. The proportions of several cell types that exhibited significant differences in cell numbers between the two groups were analyzed, as shown in Fig. 7d. Particularly striking were mast cells and macrophages (immune activation), which were almost completely lost after treatment in the control group, with their proportions reduced to only 8.2% and 1.2% compared with the device group. The difference between the two groups is also evident in the UMAP plots (Fig. S7a), where these two cell types are almost absent in the control group. Mast cells, which are known to be highly sensitive to mechanical and enzymatic stress, were 11 times more abundant in the HLS group than in the control group, where only 7 mast cells were detected. This remarkable preservation of mast cells is a testament to the gentleness and effectiveness of the HLS dissociation mechanism. Similarly, macrophages, especially immunologically activated macrophages, were more prevalent in the HLS group. The HLS group contained 80 times more immunologically activated macrophages than the control group, which retained only a single cell. This observation not only emphasizes the ability of the HLS method to minimize stress-induced immune activation but also highlights its superiority in preserving these fragile and functionally important cell types. Moreover, as renal cancer is often resistant to chemotherapy and radiotherapy, immunotherapy serves as a pivotal treatment option for advanced-stage patients. The single-cell sequencing results from the HLS group have provided a more comprehensive understanding of the renal cancer immune microenvironment, potentially aiding the development of improved therapeutic strategies for late-stage renal cancer. Extending these findings to a broader context, the ability of the HLS method to reveal the complete tumor microenvironment could greatly benefit disease management, including insights into tumor progression, therapeutic interventions, and diagnostics.

To further evaluate the impact of dissociation on cellular stress, we analyzed the expression levels of stress response genes in different cell types. A set of 140 known stress response genes was used to score various cell types, and the results were visualized through violin plots (Fig. 7e). For most cell types, including cancer cells, endothelial cells, smooth muscle cells, T/NK cells, and B cells, the stress response scores showed minimal differences between the HLS and control groups. This indicates that the non-contact, efficient dissociation method of the HLS device does not induce additional stress compared to traditional methods, further validating its mild nature. Slightly elevated stress response scores were observed in plasma cells, macrophages, and mast cells, with macrophages showing the most notable

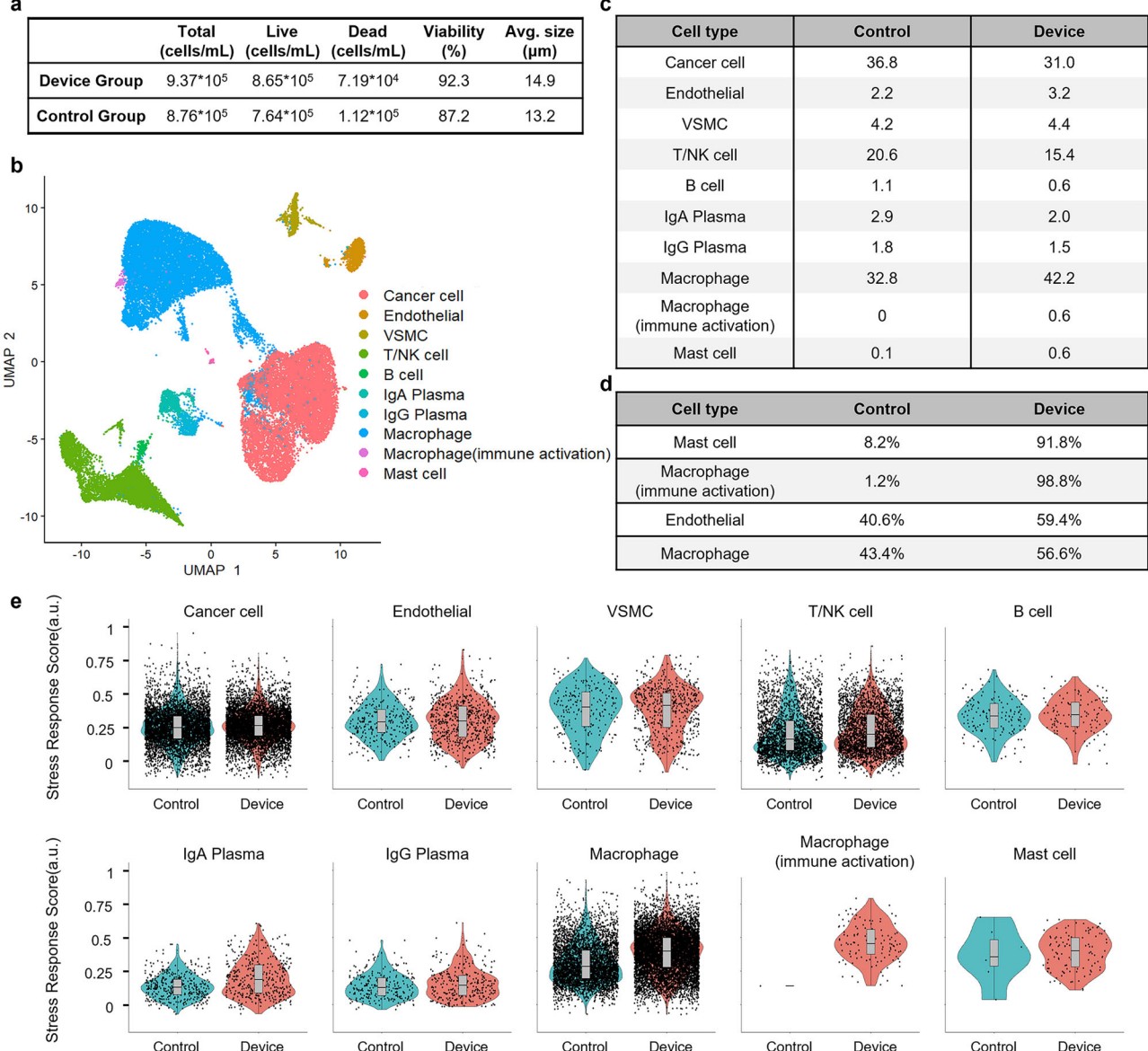

**Fig. 7 | Single-cell quality and population-level analyses after tissue dissociation.** **a** Cell quality statistics for samples prior to single-cell RNA sequencing. **b** UMAP plot showing 11 identified cell clusters. **c** Within-group population distributions of each cluster under different processing conditions. **d** Cell types with statistically significant differences in proportions between the two groups. **e** Stress response scores displayed as violin plots; horizontal lines indicate medians, and boxes represent the 1st and 3rd quartiles.

increase, albeit minor (less than 0.2 units). Despite these slight increases, the higher viability and proportion of macrophages in the HLS group suggest that the device does not compromise macrophage survival and may even enhance their retention compared to conventional methods. This finding is particularly relevant as macrophages play a crucial role in various physiological and pathological processes, and their accurate representation in single-cell analysis is essential for understanding tissue function and disease mechanisms.

In summary, the scRNA-seq analysis of samples dissociated by the HLS method has provided valuable insights into the capabilities of this contact-free dissociation technique. The HLS method not only yields a higher quantity and quality of single-cell suspensions but also demonstrates superior preservation of rare and sensitive cell types. The ability to maintain the integrity of the tissue's cellular composition and minimize cellular stress during dissociation makes the HLS method an ideal choice for single-cell analysis applications. These findings have important implications for advancing our understanding of tissue heterogeneity, disease mechanisms,

and the development of personalized medicine strategies. By providing a more accurate and comprehensive view of the cellular landscape, the HLS method may improve single-cell research and contribute to future therapeutic discovery efforts.

## Discussion
In this study, we present a contact-free tissue dissociation method based on hypersonic streaming levitation and spinning (HLS), which has yielded several important observations regarding dissociation performance and cellular outcomes. Through the use of the specially designed triple resonator probe, the HLS technique enables the levitation and noncontact treatment of tissue samples. This leads to enhanced shear forces that effectively dissociate tissues while maintaining cell integrity, increasing tissue utilization, and protecting rare cells. The HLS method, through its short-duration, non-contact mechanism, better preserves cell viability and functionality, particularly when handling rare cell populations. This is especially important for fragile cell types that are often lost in traditional dissociation methods. Our

comprehensive experiments have demonstrated the superiority of the HLS method over traditional mechanical and enzymatic dissociation methods.

One of the intriguing findings of HLS is the ability to observe and analyze the dissociation of cells over time. By varying the treatment time, we have uncovered dynamic changes in cell populations. For example, in the case of cancer cells, the proportion in the HLS (5 min) group was initially the highest at 56%, decreased to 30% in the HLS (10 min) group, and then increased again to approximately 50% in the HLS (15 min) group, slightly exceeding that of the control (40%). This pattern suggests a complex process of tissue breakdown, with the outer layers of cancer cells being preferentially dissociated first, followed by the release of deeper-lying cancer cells as the dissociation progresses. In contrast, immune cells peaked in the HLS (10 min) group at 25%, which was 2–4 times greater than that in the other groups. This indicates that the dissociation process affects the release of different cell types at different time points, providing valuable insights into the organization and interactions within the tissue.

Furthermore, the HLS method has shown remarkable preservation of rare cell types. Mast cells, which are highly sensitive to mechanical and enzymatic stress, were 11 times more abundant in the HLS group than in the control group. Similarly, immunologically activated macrophages were 80 times more prevalent in the HLS group compared to the control group. This not only emphasizes the gentleness and effectiveness of the HLS dissociation mechanism but also highlights its potential for providing a more comprehensive view of the tissue's cellular composition.

Despite the HLS platform's demonstrated advantages in dissociation efficiency, cell viability, and rare cell preservation, several limitations warrant consideration. First, the current setup is optimized for small to medium tissue volumes, and further engineering efforts are needed to enable high-throughput processing of large-scale clinical samples. Moreover, while the method has been shown to be effective for the tissue types and models tested, its applicability to a broader range of tissues and diseases remains to be fully explored. For instance, the influence of varying tissue stiffness and extracellular matrix composition on the dissociation outcome is not yet fully understood. In addition, the ability to control dissociation kinetics over time is a promising aspect of the platform that deserves further investigation. By precisely tuning acoustic parameters and enzymatic conditions, it may be possible to achieve refined, layer-by-layer dissociation. This capability could serve as a valuable tool for studying the tissue microenvironment, enabling the elucidation of spatial and temporal relationships between different cell types. For example, in cancer research, it could help reveal how tumor cells interact with surrounding stromal and immune cells at various stages of progression. In neurological disorders, it could support investigations into dynamic interactions between neurons and glial cells during disease development.

In conclusion, the HLS method represents a major advancement in tissue dissociation technology, with its unique capabilities enabling a deeper understanding of dissociation dynamics and the preservation of rare cell types. Although there are areas for improvement, the potential for future development and application in a wide range of fields is substantial, promising to enhance our understanding of biological systems and contribute to improved disease diagnosis and treatment strategies.

## Materials and methods
### Fabrication and operation of the device
The probe resonator was fabricated using standard MEMS approach as described in a previous publication[59]. In terms of thickness, the device can be divided into two parts: the Bragg mirror structure, which consists of alternating layers of silicon dioxide (SiO2) and aluminum nitride (AlN) deposited on a silicon substrate for BAW reflection, and the sandwich structure, which consists of molybdenum (Mo), AlN, and gold (Au) as the bottom electrode (BE), piezoelectric layer, and top electrode (TE) for acoustic vibration.

The automated tissue dissociation apparatus device was designed with two layers of transparent acrylic plates for easy insertion and replacement of filter membranes. It consists of four vertical tubes for inlet, sample, outlet, and waste. The sample tube has a simplified three-axis displacement stage.

The lower layer has a tapered dissociation chamber and a collection chamber for observing cell morphology. A simple XY-axis translation stage is positioned above the sample tube, securing the probe and enabling adjustment of its position within the dissociation chamber. All components are made of acrylic. The digestion chamber has a 50 degree tapered design to optimize tissue rotation. Four filter membranes with pore sizes of 5 μm, 100 μm, 40 μm, and 5 μm are used to simultaneously dissociate and filter the cells.

The enzyme solution is introduced into the system through the inlet pipe using a water pump, while the tissue is added to the dissociation chamber via the sample input pipe. The hypersonic probe is then inserted for dissociation. Once dissociation is complete, the enzyme solution is removed via the waste liquid pipe with the help of a pump. The cells are retained in the collection chamber. PBS solution is then introduced through the inlet pipe, and the cells suspended in PBS are expelled through the sample output pipe. The two chambers are specifically used for tissue dissociation and single-cell collection.

The probe consists of an RF cable, a PCB, a resonator, and an aluminum alloy casing. During assembly, the RF cable is soldered to the PCB using soldering tin. The resonator is fixed to the PCB with AB glue and connected via gold wire bonding. Finally, the aluminum alloy casing is secured with quick-drying adhesive to protect the internal components.

The probe was controlled by a sinusoidal signal (2.49 GHz), which was generated by a signal generator (Agilent, N5171B) and amplified by a power amplifier (Mini-Circuits, ZHL-5W422 + ).

### Force measurement
Force transducer (Aurora Scientific, 405 A) with a tip of 100 μm in diameter was used to characterize hypersonic streaming under different power excitation, and the value was extracted and recorded with a Source Measure Unit (Keithley 2400, USA).

### Tissue models
Human renal carcinoma tissue samples were obtained from the Department of Urology at the Second Hospital of Tianjin Medical University. After collection, the tissue samples were immediately placed in cold phosphate-buffered saline (PBS) supplemented with 1% penicillin-streptomycin (Gibco, Thermo Fisher Scientific) to prevent contamination. Before dissociation, the tissue is cut into 2 mm³ pieces.

### Finite element simulation
Simulation of the hypersonic streaming field was performed using a fluid-structure interaction model in COMSOL Multiphysics 5.5. The gigahertz vibration–induced hypersonic streaming is governed by a decaying body force generated from acoustic attenuation[63]. In the 2D simulation, the width of the body force region was adjusted to match the device area, with a height of 50 μm. The liquid was set as water, and the velocity field was described using the incompressible Navier–Stokes equations provided in COMSOL. In the 3D simulation, the body force region was shaped to match the device geometry, also with a height of 50 μm. All other parameters remained consistent with the 2D simulation.

### DNA quantification
Purified gDNA content of digested kidney tissue suspensions was assessed using a Nanodrop ND-1000 (Thermo Fisher, Waltham, MA) following isolation using a QIAamp DNA Mini Kit (Qiagen, Germantown, MD) according to the manufacturer's instructions. gDNA for device processed samples represents the cellular contents eluted from the device after operation, while gDNA for control samples represent the total amount of gDNA present in these samples.

### Tissue dissociation
Device group: 50 mg of tissue, cut into 2 mm³ pieces, is placed into the automated tissue dissociator. The device operates according to the previously mentioned procedure, with the resonator working at an average power of 3 W

for 15 min. The solutions used are 2 mg/ml collagenase II solution and PBS. Shaker group: An equal amount of tissue is placed in a centrifuge tube and 5 ml of 2 mg/ml collagenase II solution is added. The sample is then incubated at 37 °C and shaken at 200 rpm for 1 hour. The enzyme solution concentration used in all dissociation experiments was 2 mg/ml.

## Primary cell culture

Both groups of collected cell suspensions were cultured in cell culture incubators under the same conditions. Cells were grown in Dulbecco's modified Eagle medium (DMEM) supplemented with 10% fetal bovine serum and 1% penicillin–streptomycin. The cell line was maintained in T-25 cell culture flasks in an incubator at 37 °C and 5% CO2 level. Daily photography observation was carried out over a period of 11 days, with timely medium changes to ensure a conducive environment for cell growth. Every other day, cell growth was monitored at five fixed locations using an inverted microscope (Leica, DMi8) to capture images and observe the cell proliferation and morphology.

## Immunofluorescence testing

Cells were fixed with PFA and then permeabilized and blocked using Triton X-100 and 1% BSA in sequence. Primary antibodies, including mouse-sourced CA9 antibody and rabbit-sourced CD10 antibody, were applied and incubated overnight at 4 °C. The next day, secondary antibodies, anti-mouse IgG Rhodamine conjugate and anti-rabbit IgG FITC conjugate, were added and incubated in the dark at room temperature for 60 min. Finally, DAPI staining was applied for 3 min. Confocal microscopy was used to record the cell staining results.

## Flow Cytometry experiments and analysis

Cell suspensions were labeled with CA9 (renal cancer), CD45 (leukocytes), and 7-AAD (live/dead) antibodies. Samples were divided into four portions: three groups were processed with the HLS device for 5, 10, and 15 min at 3 W, while one group was processed using the shaker for 60 minutes. After staining, the samples were analyzed by flow cytometry.

Cell suspensions obtained from digested human renal carcinoma samples were stained with the fluorescent probes listed in Table S1 and analyzed using flow cytometry. Acquired data were compensated and assessed using a sequential gating scheme. Gate 1 was based on FSC-A vs SSC-A to exclude debris and cell fragments. Gate 2 selected single cells based on FSC-A vs FSC-H. Gate 3 identified immune cells based on CD45-FITC positive signal, while CA9-PE positive cells were classified as cancer cells. Gate 4 was applied to the CD45 (–)/CA9(–) cell subset to identify stromal cells. Additionally, cell viability was assessed using 7-AAD, with live cells being 7-AAD negative across cancer, immune, and stromal cell populations. Appropriate isotype controls were used to assess nonspecific background staining, and fluorescence minus one (FMO) controls were used to determine positive signal cut-offs and set gates. Control samples were left unstained to confirm baseline fluorescence.

## Single-cell RNA sequencing

The tissue was flattened and randomly divided into five cutting areas to capture cellular diversity across different spatial positions. Each area was equally partitioned for both the experimental (device) and control groups, ensuring equal tissue mass and minimizing discrepancies due to sample differences. For the experimental group, the prepared tissue samples were loaded into the automated tissue dissociator, treated at 3 W power for 15 min, and filtered through the device's internal membrane to collect pure single-cell suspensions. In the control group, an equal amount of tissue was dissociated using a shaker-based enzymatic method for 60 min, followed by filtration through a cell filter. Both single-cell suspensions were concentrated to 500 μL for cell concentration and viability analysis prior to sequencing (Fig. 7a).

The single-cell suspensions from both groups were independently processed using the droplet-based 10X Chromium platform. A total of 34,008 cells were sequenced—19,860 from the device group and 14,148 from the control group—each at an average depth of approximately 60,000 reads. After data filtering, we employed the Seurat package to identify and annotate cell clusters.

The cell suspension was loaded into Chromium microfluidic chips with chemistry and barcoded with a 10× Chromium Controller (10X Genomics). RNA from the barcoded cells was subsequently reverse-transcribed and sequencing libraries constructed with reagents from a Chromium Single Cell v2 reagent kit (10X Genomics) according to the manufacturer's instructions. Sequencing was performed with Illumina NovaSeq according to the manufacturer's instructions.

## Ethical approval

This study was approved by the Ethics Committee of The Second Hospital of Tianjin Medical University (Approval No. KY2021K095). All tissue samples were obtained from patients who underwent nephrectomy, with informed consent obtained prior to sample collection.

## Statistics

Data are represented as the mean ± standard error. Error bars represent the standard error from at least three independent experiments. $P$ values were calculated from at least three independent experiments using Student's $T$ test. Coefficient of variation was calculated as the standard error divided by the mean to represent batch-to-batch reproducibility between experimental replicates.

## Reporting summary

Further information on research design is available in the Nature Portfolio Reporting Summary linked to this article.

## Data availability

The RNA sequencing data generated in this study have been deposited in the NCBI Gene Expression Omnibus (GEO) under accession number GSE290412. The data that support the findings of this study are available from the corresponding author upon reasonable request.

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

## Acknowledgements

The authors gratefully acknowledge the financial support from the National Science Foundation of China (NSFC No.62174119, No. 22427807).

## Author contributions

Conceptualization: Y.B., X.D., and Zhihong Zhang; methodology: Y.B., Zhiwen Zheng, Z.N., and J.L.; Investigation: Y.B., X.D., Zhiwen Zheng, and Z.N.; Visualization: Y.B.; Supervision: X.D.; Writing—original draft: Y.B.; writing—review & editing: Y.B., X.D., and Zhihong Zhang.

## Competing interests

The authors declare the following competing interests: This research was funded by Convergency Biotech Ltd. (Tianjin, China), but the company did not influence the study results, which were solely the responsibility of the authors.
