## [Transparent Peer review file · Communications Engineering]

Hypersonic Levitation and Spinning: Paving the Way for Enhanced Single-Cell Analysis via Contactless Tissue Dissociation

Corresponding Author: Professor Xuexin Duan

Version 0:

Reviewer comments:

Reviewer #1

(Remarks to the Author)

This study presents an innovative Hypersonic Levitation and Spinning (HLS) tissue dissociation method, which employs a non-contact “press-and-rotate” mechanism within an automated system to achieve rapid, efficient, and gentle tissue dissociation. Compared to conventional approaches, HLS significantly enhances cell viability, preserves rare cell populations, and improves dissociation efficiency. These advantages make it particularly valuable for applications in single-cell analysis, precision medicine, and biomedical research. With robust experimental validation and broad potential impact, this work represents a significant advancement in tissue dissociation technology and a notable step forward in leveraging acoustofluidic technology to address tissue processing challenges.

I recommend this manuscript for publication after addressing the following points:

Abstract Refinement: Dividing the abstract into three paragraphs does not necessarily enhance clarity. Please condense and refine it to present key findings more concisely and effectively.

Automated Dissociation Device (Lines 138–142): Could you elaborate on the specific functions of the four-pipeline design in the automated dissociation device? How do they work together? What design rationale underlies the two chambers below? Additionally, what criteria were used in selecting the pore sizes of the four filters?

Transducer Configuration (Line 182): You mention that three transducers provide better results than a single transducer. Could you specify the advantages of this configuration? What parameters were considered in determining that three transducers were optimal?

System Flexibility (Figure 1): The study examines the effects of conical angle and probe height on flow velocity distribution. Could you demonstrate the tunability of your system by achieving stable tissue spinning and levitation using tubes with 30° and 70° angles, given an optimized probe position? This would help establish the platform’s universality.

Jet Forces in Tissue Dissociation (Line 262): How do jet forces effectively drive tissue dissociation without inducing cell damage, particularly within the 10–70 μN range? Are these forces directly applied to the tissues?

Control Group Selection (Line 310): The study uses the traditional shaker method for one hour as a control. Given that dissociation time is critical for preserving cellular integrity, would a 15-minute comparison group provide a more appropriate benchmark?

Figure Labeling (Lines 349–352): For better clarity, should the a), b), c), d) labels be placed before the corresponding descriptive text?

Cell Counting in Figure 4b (Line 395): In Figure 4b, why does the cell count remain in the hundreds? How was cell counting performed to characterize growth dynamics during cell culture? Please provide a detailed methodology.

Primary Culture Cycle Duration (Line 397): Is the primary culture cycle duration one day or 11 days? Please clarify.

Stress-Induced Immune Activation (Line 585): How does the HLS method minimize stress-induced immune activation while simultaneously increasing the presence of macrophages and mast cells?

Stress-Response Gene Selection (Lines 596–598): What criteria were used to select the 140 stress-response genes? How was the stress-response score calculated?

Relevant References: I suggest incorporating the following references, which focus on the design of acoustofluidic systems for single-cell processing and analysis. These works provide valuable insights into acoustic streaming, microfluidics, and single-cell manipulation, strengthening the manuscript by situating the HLS method within the broader context of acoustofluidic-based single-cell technologies and highlighting its advancements over existing approaches:

'Harmonic acoustics for dynamic and selective particle manipulation', *Nature Materials*, Vol. 21, pp. 540-546, 2022.

'Acoustic tweezers for high-throughput single-cell analysis', *Nature Protocols*, 18(8), 2441-2458, 2023.

'Acoustofluidics for biomedical applications', *Nature Reviews Methods Primers*, Vol. 2, p. 30, 2022.

'Quantifying cell adhesion through forces generated by acoustic streaming', *Ultrasonics Sonochemistry*, Vol. 90, 106204, 2022.

'Acoustofluidic large-scale mixing for enhanced microfluidic immunostaining for tissue diagnostics', *Lab Chip*, Vol. 23, pp. 3258-3271, 2023.

Addressing these points will further strengthen the manuscript's clarity, rigor, and impact. I look forward to seeing these revisions incorporated.

Reviewer #2

(Remarks to the Author)

The experimental results of this study demonstrate the advantages of the HLS dissociation method; however, several critical points remain unclear, as outlined below. Based on these concerns, I conclude that the manuscript is not suitable for publication in its current form.

1. The principle of HLS dissociation is schematically illustrated in Scheme 1a and Fig. S1. The authors consider F_r (which must not be interpreted as "pressure") as the primary factor driving HLS dissociation. However, if this assumption holds, the tissue would not remain stationary at a specific position during the process, as force influences both deformation and displacement. Given this, the tissue is likely affected solely by rotational flow, meaning it becomes trapped at the stagnation point of the rotational motion. Consequently, centripetal force may serve as a more efficient and productive net force, facilitating deeper enzyme penetration into the tissue.

In addition, the authors claim that rapid self-rotation amplifies the shear stress on tissue surfaces. However, is this assumption valid? If the tissue rotates in a stationary medium, high-speed rotation would indeed induce significant shear stress. However, in the HLS system, shear stress is likely to be minimal, as the tissue rotates in accordance with the rotational flow rather than experiencing relative motion against the surrounding medium.

I recommend that the authors reconsider the dissociation mechanism.

2. When comparing HLS dissociation with the conventional method, as shown in Scheme 1d, the authors state that HLS dissociation applies a higher force (kinetic stimulation) than the conventional approach. Since both methods utilize enzymes, what then is the primary factor responsible for maintaining functional integrity and cell viability? The experimental results may provide evidence for cell quality; however, the underlying hypothesis remains questionable.

3. Fig. 1 does not contain a tissue, or an elastic object. However, the presence of an object in a viscous fluid can influence the flow distribution. The discussion presented here appears overly optimistic or may lead to potential misinterpretation.

4. L. 167: The authors refer to the acoustic pressure field; however, Fig. 1a appears to illustrate the velocity distribution, does it not?

5. I could not find a correspondence in the enzyme conditions among the HLS, shaker, and enzyme-only experiments.

6. The proposed method must be compared with the traditional shaker method at the same time point to ensure a valid comparison. It is unclear whether the gDNA trend in Fig. 3a follows a linear relationship over time or not.

7. Fig. 3 needs statistical analysis to conclude the difference of "enzyme only", "no enzyme + shaker," and "no enzyme + device."

8. How did you obtain the results shown in Fig. 3d? The vertical axis represents "Cell count/min". In most cases, there is a monotonically decreasing trend. Does it mean the total number of cells is almost the same, but the process time (denominator) increases? Or the cell count per unit time at each time point?

9. From Fig. 4a and Figs. S3 and S4, the difference seems to be due to the difference in initial cell number. It is an overstatement that "the HLS device group exhibited superior growth performance compared to the shaker group" in Ls. 393-394. The growth rate illustrated in Fig. 4c does not have a big difference.

10. The statement in Line 400, "the gentler dissociation method employed by the HLS device," is questionable. If the method is truly gentler, why does dissociation occur faster? A harsher process would logically lead to shorter dissociation times, thereby reducing enzymatic exposure and preserving surface proteins, which, in turn, would facilitate faster initial adhesion. Similarly, the statement in Line 405, "the HLS dissociation device causes less damage to cells," appears to be an

overstatement. I recommend that the authors avoid using the term “damage” in this context. Instead, “preserving surface proteins/antigens” would be a more appropriate and precise description.

One concern regarding Fig. 4c is that the growth rate for the shaker group appears more stable across cycles. Notably, could the first cycle in the device group be considered an outlier, as it may not accurately reflect the normal state of the cells? Immunofluorescence staining in Fig. 4d should be compared across groups and cycles to ensure a comprehensive evaluation.

11. The graph legends should clearly indicate the meaning of each plot and error bars. Although the “Statistics” section provides an explanation, the number of trials (n) likely varies across experiments and should be explicitly stated for each case.

Additionally, there is no indication of statistical significance in the graphs. The authors mention that Student’s t-test was used for analysis; however, in some cases, analysis of variance (ANOVA) would be more appropriate for comparing multiple groups. I recommend reconsidering the statistical approach to ensure the validity of the analysis.

12. The abstract lacks concrete results from this study. For instance, the lack of information on cell type, and how accurate are the following assays compared with the traditional approaches.

13. Ls. 227, 228: The unit “K um²” looks strange to me.

14. Fig. 2a requires additional descriptions for each component to enhance clarity and understanding. The enlarged figures should be aligned to correspond in angle for consistency and accurate comparison.

15. L. 450: “30%” should be corrected to “33%”. Additionally, the values for HLS and control should be accurately stated as they appear in Fig. 5.

Reviewer #4

(Remarks to the Author)

In the manuscript “Hypersonic Levitation and Spinning: Paving the Way for Enhanced Single-Cell Analysis via Contactless Tissue Dissociation,” Bai et al introduced a novel tissue dissociation method based on hypersonic levitation and spinning (HLS), claiming that it represents a breakthrough in single-cell analysis. Essentially, the device employs a specially designed triple-acoustic resonator probe to levitate and spin the tissue, applying precise hydrodynamic shear forces without any direct contact. This non-contact approach dramatically reduces the time required for dissociation—achieving a 90% tissue utilization rate in just 15 minutes—while better preserving cell viability and rare cell populations compared to traditional mechanical or enzymatic methods.

The authors support their claims with a robust set of experimental data. For example, they demonstrate that the HLS method recovers nearly 90% of gDNA compared to controls, while a conventional shaker-based method only reaches about 70% after one hour. Additionally, primary cell culture experiments indicate that cells processed with HLS adhere and proliferate more effectively in the early stages. Flow cytometry and single-cell RNA sequencing further confirm that HLS preserves the tissue’s original cellular composition more faithfully, including the maintenance of delicate cell types such as mast cells and activated macrophages.

A significant strength of this work lies in the combination of simulation and experimental validation. Finite element simulations were used to optimize critical parameters—such as the cone angle and resonator height—that are essential for creating an optimal flow field for tissue dissociation. This careful engineering ensures that the device applies uniform and controlled forces, resulting in a gentler dissociation process that minimizes cell damage.

However, the paper could be improved by:

1. Side-by-side comparisons of UMAP visualizations and key metrics (such as genes per cell and mitochondrial content per cell) for single-cell RNA sequencing data from both the traditional dissociation method and HLS would provide clearer insights into performance differences.
2. Since the study primarily focuses on kidney tissue—a relatively straightforward sample to dissociate—the added value of HLS appears limited except for the detection of a new type of activated macrophage. A brief testing the device on other tissue types, such as brain tissue, would better demonstrate its general applicability.

Overall, the HLS method shows great potential to transform tissue dissociation, particularly in fields that rely on single-cell analysis. Its rapid and gentle processing could lead to improved diagnostic outcomes and more accurate insights into cellular heterogeneity, which are crucial for advancing cancer research and personalized medicine. With further validation across a broader range of tissues and more detailed statistical analysis, this innovative approach could reshape current tissue dissociation practices and have a significant impact on both basic research and clinical applications.

Version 1:

Reviewer comments:

Reviewer #1

(Remarks to the Author)

I appreciate the authors' efforts in revising the manuscript. They have adequately addressed the concerns raised during the first round of review, and the overall quality of the manuscript has improved significantly. I believe the paper is now suitable for publication.

I have only two minor suggestions for further polishing:

1. Add Scale Bars to All Panels in Figure 4: Currently, only the last panel includes a scale bar. For clarity and consistency, I recommend adding scale bars to all subfigures in Figure 4.
2. Discuss Potential Limitations: While the manuscript presents a promising platform for tissue dissociation, it would benefit from a brief discussion of potential limitations. This addition would provide a more balanced and comprehensive view of the platform's capabilities and practical applicability.

Overall, I recommend acceptance, with minor edits to improve clarity and completeness.

Reviewer #2

(Remarks to the Author)

Thank you for addressing my review comments in the revised manuscript. Your responses have clarified several points and enhanced my understanding. I find the current version of the manuscript satisfactory.

Reviewer #4

(Remarks to the Author)

The authors have addressed both of my questions. I do not have further concerns.

Response to Reviewer 1:

Dear Reviewer,

Thank you for your valuable comments on our research. In response to your various points, we have made detailed revisions and added supplementary explanations. Below is our detailed response to each of your comments.

1. Abstract Refinement: Dividing the abstract into three paragraphs does not necessarily enhance clarity. Please condense and refine it to present key findings more concisely and effectively.

Response: Thank you for your suggestion. We have added more specific experimental results in the abstract to present the core findings of the study more clearly. We included the results showing that the HLS method achieves a tissue utilization rate of 90% and a cell viability of 92.3% within 15 minutes, clearly demonstrating its advantages in efficiency and cell retention.

2. Automated Dissociation Device (Lines 138–142): Could you elaborate on the specific functions of the four-pipeline design in the automated dissociation device? How do they work together? What design rationale underlies the two chambers below? Additionally, what criteria were used in selecting the pore sizes of the four filters?

Response: Thank you for your question. The four pipes are used for fluid intake, sample introduction, fluid output, and waste liquid discharge, enabling automated liquid exchange and filtration. The enzyme solution is introduced into the system through the inlet pipe using a water pump, while the tissue is added to the dissociation chamber via the sample input pipe. The hypersonic probe is then inserted for dissociation. Once dissociation is complete, the enzyme solution is removed via the waste liquid pipe with the help of a pump. The cells are retained in the collection chamber. PBS solution is then introduced through the inlet pipe, and the cells suspended in PBS are expelled through the sample output pipe. The two chambers are specifically used for tissue dissociation and single-cell collection.

Regarding the filters, we selected pore sizes of 100 μ m, 70 μ m, 40 μ m, and 20 μ m to achieve efficient stepwise filtration from tissue fragments to single cells, based on the size distribution of major cell types in human solid tumors.

To clarify these issues, we have incorporated the revised workflow of the device into the Methods section (lines 725–731).

3. Transducer Configuration (Line 182): You mention that three transducers provide better results than a single transducer. Could you specify the advantages of this configuration? What parameters were considered in determining that three transducers were optimal?

Response: Thank you for your question. The triple-resonator design allows the tissue to experience stable eccentric thrust (F_r) and rotational shear forces (F_s) within the acoustic field, leading to stable, high-speed rotation. A single resonator can only provide bias or local disturbance, making it difficult to form an overall flow field envelope. Through simulations (Figure 1a, b), we have validated that the triple-resonator configuration outperforms the single resonator in both flow field stability and flow velocity.

4. System Flexibility (Figure 1): The study examines the effects of conical angle and probe height on flow velocity distribution. Could you demonstrate the tunability of your system by achieving stable tissue spinning and levitation using tubes with 30° and 70° angles, given an optimized probe position? This would help establish the platform's universality.

Response: Thank you for your question. In Figure 1, we conducted simulations to verify the probe position, height, and container cone angle, with the aim of optimizing parameters for subsequent experiments. From our simulations, we observed that a 50° cone angle allows stable dissociation for tissue sizes ranging from 1 to 3 cubic millimeters. At this tissue size, cone angles of 30° and 70° were unable to consistently achieve stable tissue rotation and levitation. Considering the stability and reproducibility of the data for subsequent experiments, we selected a 50° cone angle for the experiments. We also conducted simulations to assess the spatial positioning of the probe, including its height and lateral placement. As shown in Figure 1, adopting the optimized spatial configuration significantly contributes to the stability and efficiency of the dissociation process.

5. Jet Forces in Tissue Dissociation (Line 262): How do jet forces effectively drive tissue dissociation without inducing cell damage, particularly within the 10–70 μN range? Are these forces directly applied to the tissues?

Response:

Thank you for your question. The jet forces in the range of 10–70 μN are applied to the tissue surface in the form of shear stress within the liquid medium. These forces are distributed widely and evenly, without directly acting on individual cells. As reported in “Adv. Sci. 8, 2002489 (2020)”, forces in the range of several micronewtons were also used to directly compress cells without causing damage. We have further validated the integrity of the cell membrane using flow cytometry and immunofluorescence (Figures 5 and 4d), demonstrating that the cell membrane remains intact.

6. Control Group Selection (Line 310): The study uses the traditional shaker method for one hour as a control. Given that dissociation time is critical for preserving cellular integrity, would a 15-minute comparison group provide a more appropriate benchmark?

Response: Thank you for your suggestion. We conducted the gDNA experiment for the shaker 15-minute group and have added the data in the manuscript, as shown in Figure 3. We also clarified that, at the 15-minute mark, the tissue block size in the shaker group did not show significant changes visually, and the dissociation effect was poor. Therefore, for subsequent dissociation experiments, the traditional shaker method was used with a 1-hour treatment time as the control.

7. Figure Labeling (Lines 349–352): For better clarity, should the a), b), c), d) labels be placed before the corresponding descriptive text?

Response: Thank you for your suggestion. I have moved the labels to precede the descriptive text.

8. Cell Counting in Figure 4b (Line 395): In Figure 4b, why does the cell count remain in the hundreds? How was cell counting performed to characterize growth dynamics during cell culture? Please provide a detailed methodology.

Response: The cell count in this case was obtained by counting cells in five fixed fields of view using an inverted microscope and averaging the results. This method differs from direct flow cytometry counting and has been specified in the Methods section. During the cell growth process, images were taken for 11 days at five randomly selected sites (marked locations) using the inverted microscope.

9. Primary Culture Cycle Duration (Line 397): Is the primary culture cycle duration one day or 11 days? Please clarify.

Response: Thank you for your question. We have corrected the relevant description. The complete primary culture lasted for 11 days, with each cycle spanning 2 days, totaling 5 cycles. The images shown in the figure represent daily samples (Figure 4a), and the statistical data are based on measurements taken every 2 days (Figures 4b-c).

10. Stress-Induced Immune Activation (Line 585): How does the HLS method minimize stress-induced immune activation while simultaneously increasing the presence of macrophages and mast cells?

Response: Thank you for your question. The HLS method, through its non-contact, short-duration shear mechanism, effectively separates immune cells from tissue, such as macrophages and mast cells. In traditional dissociation methods, cells typically undergo prolonged exposure to enzymatic solutions and mechanical forces, which can lead to cell membrane damage, loss of surface antigens, and immune activation. In contrast, the HLS method significantly shortens the dissociation time, reducing cell exposure to enzymatic solutions, thereby avoiding the common stress responses observed in traditional methods. During dissociation, the cells are not subjected to excessive physical stress, so their functionality is better preserved. This advantage is especially evident when handling fragile immune cells, where the HLS method shows clear superiority. Experimental results indicate that the HLS method not only significantly improves the survival rate of macrophages and mast cells but also protects their immune function by minimizing unnecessary immune activation.

11. Stress-Response Gene Selection (Lines 596–598): What criteria were used to select the 140 stress-response genes? How was the stress-response score calculated?

Response: The 140 genes used in this study are sourced from publicly available cell stress

response databases. The detailed genes and calculation methods follow the approach described in Lombardo's paper. The stress response gene score was calculated using the AddModuleScore function in Seurat.

Reference:

Lombardo, J.A. et al. Microfluidic platform accelerates tissue processing into single cells for molecular analysis and primary culture models. Nat Commun 12, 2858 (2021).

12. Relevant References: I suggest incorporating the following references, which focus on the design of acoustofluidic systems for single-cell processing and analysis. These works provide valuable insights into acoustic streaming, microfluidics, and single-cell manipulation, strengthening the manuscript by situating the HLS method within the broader context of acoustofluidic-based single-cell technologies and highlighting its advancements over existing approaches:

'Harmonic acoustics for dynamic and selective particle manipulation', Nature Materials, Vol. 21, pp. 540-546, 2022.

'Acoustic tweezers for high-throughput single-cell analysis', Nature Protocols, 18(8), 2441-2458, 2023.

'Acoustofluidics for biomedical applications', Nature Reviews Methods Primers, Vol. 2, p. 30, 2022.

'Quantifying cell adhesion through forces generated by acoustic streaming', Ultrasonics Sonochemistry, Vol. 90, 106204, 2022.

'Acoustofluidic large-scale mixing for enhanced microfluidic immunostaining for tissue diagnostics', Lab Chip, Vol. 23, pp. 3258-3271, 2023.

Addressing these points will further strengthen the manuscript's clarity, rigor, and impact. I look forward to seeing these revisions incorporated.

Response: Thank you for providing the reference. I have carefully read it and cited it in the manuscript.

Response to Reviewer 2:

Dear Reviewer,

Thank you for your valuable comments on our research. In response to your various points, we have made detailed revisions and added supplementary explanations. Below is our detailed response to each of your comments.

1. The principle of HLS dissociation is schematically illustrated in Scheme 1a and Fig. S1. The authors consider F_r (which must not be interpreted as “pressure”) as the primary factor driving HLS dissociation. However, if this assumption holds, the tissue would not remain stationary at a specific position during the process, as force influences both deformation and displacement. Given this, the tissue is likely affected solely by rotational flow, meaning it becomes trapped at the stagnation point of the rotational motion. Consequently, centripetal force may serve as a more efficient and productive net force, facilitating deeper enzyme penetration into the tissue.

In addition, the authors claim that rapid self-rotation amplifies the shear stress on tissue surfaces. However, is this assumption valid? If the tissue rotates in a stationary medium, high-speed rotation would indeed induce significant shear stress. However, in the HLS system, shear stress is likely to be minimal, as the tissue rotates in accordance with the rotational flow rather than experiencing relative motion against the surrounding medium.

I recommend that the authors reconsider the dissociation mechanism.

Response:

Thank you for your insightful comment. We agree that the mechanical processes involved in HLS dissociation are complex and multifactorial, and we acknowledge that your point regarding the possible role of centripetal force is well-reasoned and worth consideration.

The principle of HLS dissociation is based on the generation of a microscale, controllable rotational flow field. Due to the existence of a velocity gradient, this flow field has a central equilibrium point. Tissue fragments of appropriate size become stably trapped in this central region and undergo self-rotation. During this process, the tissue is subjected to multiple forces: an impulsive acoustic streaming thrust (F_r), a rotational shear force (F_s) caused by the surrounding vortex, and also a centripetal component arising from the velocity gradient.

These forces act synergistically to facilitate deeper enzyme penetration into the tissue. As observed during dissociation, larger tissue masses initially exhibit stable self-rotation. As enzymes progressively infiltrate and digest internal structures, the larger fragments break down into smaller pieces, which then tend to follow the rotational stream more passively.

We acknowledge that this dynamic force interaction is difficult to measure directly and may involve a combination of pressure, shear, and centripetal forces. Together, these contribute to the acceleration of the tissue dissociation process. In response to your comment, we have revised the section explaining the dissociation mechanism in the manuscript (lines 188-196) to reflect this more nuanced and physically comprehensive perspective.

Additionally, we have revised the description of "rapid spinning amplifies the shear stress on the tissue surface." We further explain that, compared to translational dissociation, the shear force generated by tissue spin dissociation is greater, which facilitates enzyme penetration and accelerates the detachment of dissociated material. The modified content

has been added to lines 284–290 of the manuscript, where we clarify that, unlike the translational dissociation method, where tissue dissociates as it moves with the fluid flow, more of the fluid energy is utilized to rotate the tissue rather than allowing it to move with the fluid. In the HLS system, tissue spin generates stronger shear forces. This amplified shear force, resulting from the tissue's rotation around its own axis, is more effective at stripping dissociated material, disrupting cell-cell and cell-matrix connections, and thus accelerating the enzymatic digestion process.

Reference:

Zhang, J. et al. Surface acoustic waves enable rotational manipulation of *Caenorhabditis elegans*. *Lab Chip* 19, 984–992 (2019).

2. When comparing HLS dissociation with the conventional method, as shown in Scheme 1d, the authors state that HLS dissociation applies a higher force (kinetic stimulation) than the conventional approach. Since both methods utilize enzymes, what then is the primary factor responsible for maintaining functional integrity and cell viability? The experimental results may provide evidence for cell quality; however, the underlying hypothesis remains questionable.

Response:

Thank you very much for your question. While both the HLS and conventional methods rely on enzymatic digestion, the key distinction lies in the mechanical environment in which the enzymes act. We have further clarified the theoretical advantages of the HLS method in preserving cell functionality and viability:

1. Short-duration, non-contact microscale rotational flow: In conventional methods such as shaking or manual dissociation, shear forces are generated by macroscopic fluid turbulence and tissue collisions, typically over prolonged durations. This increases the risk of physical damage to the cell membrane, alteration of membrane protein structures, and induction of stress responses. In contrast, the HLS method utilizes localized microscale rotational flow to apply a combination of controlled mechanical forces to the tissue, enhancing enzyme penetration into deeper regions. This approach significantly shortens the dissociation time and reduces cumulative enzymatic exposure, thereby lowering the likelihood of damage to the cell membrane and internal structures.

2. More precise and energy-focused fluid-mediated dissociation: From an energy perspective, the microscale rotational flow generated in the HLS system delivers concentrated mechanical energy directly to the target region. This localized, fluid-mediated mechanism improves the spatial precision and efficiency of enzymatic activity, minimizing off-target effects and mechanical disruption while promoting deeper enzymatic infiltration. Compared to traditional agitation methods that distribute mechanical energy broadly and diffusely, the HLS flow pattern is more focused and targeted, resulting in more effective yet gentler tissue breakdown.

In addition, we have revised our terminology in the manuscript. Specifically, we removed the descriptor “mild” and “gentle” previously used for the HLS method, and replaced it with “efficient”.

We have incorporated these theoretical clarifications into the revised manuscript in the “Discussion” section (lines 662–665), highlighting that the HLS method, through its short-duration, non-contact shear mechanism, better preserves cell viability and functionality,

particularly in the dissociation of rare and sensitive cell populations.

3. Fig. 1 does not contain a tissue, or an elastic object. However, the presence of an object in a viscous fluid can influence the flow distribution. The discussion presented here appears overly optimistic or may lead to potential misinterpretation.

Response: Thank you for your valuable suggestion. In the revised manuscript, we have further clarified the simulation in Figure 1, explicitly stating that the flow field simulation in Figure 1 is based on a simplified model. To simplify the numerical calculations, we did not include elastic materials in Figure 1a and 1b. In this model, the flow field simulation does not fully reflect the real tissue dissociation process but rather serves as a reference for setting physical parameters for the actual experiments. The simulation was not intended to explain the complete physical flow phenomenon but rather to select appropriate device positioning, height, cone angles, and flow field configuration parameters. We have used the simulation to provide theoretical guidance for the experimental design, and the experimental results (e.g., Movie S1) confirm that the tissue is indeed stably captured and spun, effectively dissociating.

These additional explanations have been updated in lines 169-172 of the manuscript.

4. L. 167: The authors refer to the acoustic pressure field; however, Fig. 1a appears to illustrate the velocity distribution, does it not?

Response: We sincerely apologize for not accurately expressing this in the initial submission. Figure 1a indeed shows the velocity vector and velocity magnitude distribution, not the acoustic pressure field. In the revised manuscript, we have changed the description to "hypersonic streaming velocity field" (line 173) and accordingly adjusted the figure legend. Thank you for your careful attention to this matter.

5. I could not find a correspondence in the enzyme conditions among the HLS, shaker, and enzyme-only experiments.

Response: We have further clarified the experimental details in the "Methods" section (lines 777-778). All experimental groups were treated with collagenase II solution at the same concentration (2 mg/mL), and the solution was prepared in the same PBS buffer.

6. The proposed method must be compared with the traditional shaker method at the same time point to ensure a valid comparison. It is unclear whether the gDNA trend in Fig. 3a follows a linear relationship over time or not.

Response: Thank you for your suggestion. We conducted the gDNA experiment for the shaker 15-minute group and have added the data in the manuscript, as shown in Figure 3. Additionally, we clarified that, at the 15-minute mark, the tissue block size in the shaker group did not show significant changes visually, and the dissociation effect was poor.

7. Fig. 3 needs statistical analysis to conclude the difference of “enzyme only”, “no enzyme + shaker,” and “no enzyme + device.”

Response: We place great importance on statistical analysis and have performed three independent replicates (n=3) for the data in Figure 3. We used Two-sided T test to test for differences between the groups. All data points with statistically significant differences have been marked with significance symbols ($p < 0.05$, $p < 0.01$), and non-significant points are labeled as “ns.” The detailed statistical methods have been explained in the "Methods" section.

8. How did you obtain the results shown in Fig. 3d? The vertical axis represents “Cell count/min”. In most cases, there is a monotonically decreasing trend. Does it mean the total number of cells is almost the same, but the process time (denominator) increases? Or the cell count per unit time at each time point?

Response: Thank you for your question. The "cell count/minute" represented on the y-axis of Figure 3d is derived from the derivative of the data in Figure 3c, representing the real-time dissociation rate per minute. This data is used to analyze the differences in dissociation rates between the groups, helping to further understand the dissociation efficiency across the different groups.

9. From Fig. 4a and Figs. S3 and S4, the difference seems to be due to the difference in initial cell number. It is an overstatement that “the HLS device group exhibited superior growth performance compared to the shaker group” in Ls. 393-394. The growth rate illustrated in Fig. 4c does not have a big difference.

Response: Thank you for raising the important issue. In the revision, we have added an explanation regarding the differences in the initial cell count. The purpose of this experiment is not to demonstrate that HLS dissociation results in better cell growth, but rather to show that HLS retains more cells after dissociation, and that the acoustic streaming stimulus has no effect on cell growth. Since HLS is more effective at separating cells from tissue, the initial cell count is higher than that in traditional methods. We have modified the description of the growth rate in Figure 4c (lines 415-416) and replaced "better growth performance" with "more initial cell adhesion and slightly higher overall proliferative efficiency" to accurately describe the differences in cell growth between the two groups.

10. The statement in Line 400, “the gentler dissociation method employed by the HLS device,” is questionable. If the method is truly gentler, why does dissociation occur faster? A harsher process would logically lead to shorter dissociation times, thereby reducing enzymatic exposure and preserving surface proteins, which, in turn, would facilitate faster initial adhesion. Similarly, the statement in Line 405, “the HLS dissociation device causes less damage to cells,” appears to be an overstatement. I recommend that the authors avoid using the term “damage” in this context. Instead, “preserving surface proteins/antigens” would be a more appropriate and precise

description.

One concern regarding Fig. 4c is that the growth rate for the shaker group appears more stable across cycles. Notably, could the first cycle in the device group be considered an outlier, as it may not accurately reflect the normal state of the cells? Immunofluorescence staining in Fig. 4d should be compared across groups and cycles to ensure a comprehensive evaluation.

Response: Thank you for your detailed feedback. We have replaced "less damage" in line 428 with "preserves cell surface antigens and proteins," which more precisely expresses that the HLS method preserves the integrity of cell membrane proteins through non-contact, short-duration shear forces, thereby enhancing cell functionality and viability.

Regarding the faster cell growth rate observed in the first cycle of the device group in Fig. 4c, it is likely due to fact that the initial adaptation of freshly dissociated cells to the in vitro culture environment. At the same time, this enhanced proliferation in first cycle also suggests that the dissociation process preserved cell viability and proliferative capacity, further supporting the gentle nature of the method. As the culture progressed, the growth trends in the device and shaker groups became comparable from cycle 2 to 4, indicating that overall cell proliferation was maintained.

We agree that it is important to verify whether this value represents a statistical outlier or a biologically meaningful observation. To address this, we performed two standard statistical tests for outlier detection:

We calculated the Z-score of the first cycle's growth rate (mean = 181) relative to the mean and standard deviation of all five cycles. $Z = \frac{181-151}{22.16} \approx 1.35$. Since $|Z| < 2$, this value does not meet the threshold for statistical outlier detection.

In addition, following the reviewer's suggestion, we have included immunofluorescence staining images for the control group (see Supplementary Figure S6), comparing the expression of key surface markers across different passages. The results show that surface proteins/antigens were well preserved in cells dissociated using our device, further supporting the conclusion that the dissociation process had minimal impact on cell surface integrity and function.

11. The graph legends should clearly indicate the meaning of each plot and error bars. Although the "Statistics" section provides an explanation, the number of trials (n) likely varies across experiments and should be explicitly stated for each case.

Additionally, there is no indication of statistical significance in the graphs. The authors mention that Student's t-test was used for analysis; however, in some cases, analysis of variance (ANOVA) would be more appropriate for comparing multiple groups. I recommend reconsidering the statistical approach to ensure the validity of the analysis.

Response: Thank you for your valuable comments on the statistical analysis section. We have made further modifications and improvements to the error bars and statistical analysis methods in the charts.

12. The abstract lacks concrete results from this study. For instance, the lack of information on cell type, and how accurate are the following assays compared with the traditional approaches.

Response: Thank you for your suggestion. We have added more specific experimental results in the abstract to present the core findings of the study more clearly. We included the results showing that the HLS method achieves a tissue utilization rate of 90% and a cell viability of 92.3% within 15 minutes, clearly demonstrating its advantages in efficiency and cell retention. We also particularly highlighted the significant advantage of the HLS method in preserving rare cells, such as mast cells, with a retention rate more than 10 times higher than that of traditional methods. Additionally, we included a comparison with traditional dissociation methods (such as the shaker method), showing that the HLS method achieves higher cell retention and viability in a shorter time.

13. Ls. 227, 228: The unit “K μm^2 ” looks strange to me.

Response: Thank you for pointing out this issue. We have corrected the unit "kum2" to μm^2 (lines 242-243), which is used to represent the area unit of the transducer. All units mentioned in the manuscript have been reviewed and standardized for consistency.

14. Fig. 2a requires additional descriptions for each component to enhance clarity and understanding. The enlarged figures should be aligned to correspond in angle for consistency and accurate comparison.

Response: Thank you for your suggestion. We have revised Figure 2a. Each component in Figure 2a has been clearly labeled with text, and additional necessary explanations have been added to enhance the clarity and understanding of the diagram.

15. L. 450: “30%” should be corrected to “33%”. Additionally, the values for HLS and control should be accurately stated as they appear in Fig. 5.

Response: Thank you for your careful attention. We have made the correction as per your suggestion, changing 30% to 33%(line 475). All percentage data mentioned in the manuscript has been reviewed and corrected to 33% to ensure consistency with the data in Figure 5.

Response to Reviewer 4:

Dear Reviewer,

Thank you for your valuable comments on our research. In response to your various points, we have made detailed revisions and added supplementary explanations. Below is our detailed response to each of your comments.

1. Side-by-side comparisons of UMAP visualizations and key metrics (such as genes per cell and mitochondrial content per cell) for single-cell RNA sequencing data from both the traditional dissociation method and HLS would provide clearer insights into performance differences.

Response: Thank you for your suggestion. We agree that side-by-side comparisons of UMAP visualizations, as well as key metrics such as genes per cell and mitochondrial content per cell, would provide clearer insights into the performance differences between the traditional dissociation method and HLS. These results are provided in Supplementary Figure S7, including a side-by-side comparison of UMAP plots, gene counts per cell, and mitochondrial gene content per cell between the device group (HLS method) and the control group (traditional dissociation method). These additional data clearly highlight the enhanced performance of the HLS method in terms of cell viability, recovery, and overall dissociation efficiency, which are critical for downstream single-cell analysis.

2. Since the study primarily focuses on kidney tissue—a relatively straightforward sample to dissociate—the added value of HLS appears limited except for the detection of a new type of activated macrophage. A brief testing the device on other tissue types, such as brain tissue, would better demonstrate its general applicability.

Response: Thank you for your feedback. While this study primarily focuses on kidney tissue, we believe the results still hold significant value due to the clear advantages of the HLS method in preserving cell viability and rare cell populations. However, we also acknowledge that testing on other tissue types, such as brain tissue, would better demonstrate the general applicability of the device. We have included dissociation experiments for mouse kidney and lung tissues to demonstrate its dissociation capability for other tissues. The data from this experiment are presented in tabular form in Supplementary Figure S8. We plan to include additional tissue types in future experiments to further validate the device's broad applicability.

Response to Reviewer 1:

Dear Reviewer,

Thank you for your valuable comments on our research. In response to your various points, we have made detailed revisions and added supplementary explanations. Below is our detailed response to each of your comments.

1. Add Scale Bars to All Panels in Figure 4: Currently, only the last panel includes a scale bar. For clarity and consistency, I recommend adding scale bars to all subfigures in Figure 4.

We appreciate the reviewer's helpful suggestion. In the revised Figure 4, we have added scale bars to all panels to ensure clarity and consistency. The figure caption has also been updated accordingly.

2. Discuss Potential Limitations: While the manuscript presents a promising platform for tissue dissociation, it would benefit from a brief discussion of potential limitations. This addition would provide a more balanced and comprehensive view of the platform's capabilities and practical applicability.

We appreciate the reviewer's thoughtful suggestion. In response, we have added a dedicated paragraph in the Discussion section (now beginning with "Despite the HLS platform's demonstrated advantages...") to acknowledge and elaborate on the potential limitations of the HLS method. Specifically, we discuss the current throughput limitation for large-volume clinical samples, the need for further validation across tissue types with varying mechanical and compositional properties, and the operational complexity of tuning dissociation dynamics over time. We also outline future directions, including acoustic and enzymatic optimization for layer-by-layer detachment and exploration of spatial-temporal cellular interactions in various biological contexts. We believe this addition provides a more balanced and realistic perspective on the system's capabilities and future development.